# Deep Learning-Assisted Design of Novel Promoters in *Escherichia coli*

*Xinglong Wang, Kangjie Xu, Yameng Tan, Shangyang Yu, Xinyi Zhao, and Jingwen Zhou\**

Deep learning (DL) approaches have the ability to accurately recognize promoter regions and predict their strength. Here, the potential for controllably designing active *Escherichia coli* promoter is explored by combining multiple deep learning models. First, "DRSAdesign," which relies on a diffusion model to generate different types of novel promoters is created, followed by predicting whether they are real or fake and strength. Experimental validation showed that 45 out of 50 generated promoters are active with high diversity, but most promoters have relatively low activity. Next, "Ndesign," which relies on generating random sequences carrying functional −35 and −10 motifs of the sigma70 promoter is introduced, and their strength is predicted using the designed DL model. The DL model is trained and validated using 200 and 50 generated promoters, and displays Pearson correlation coefficients of 0.49 and 0.43, respectively. Taking advantage of the DL models developed in this work, possible 6-mers are predicted as key functional motifs of the sigma70 promoter, suggesting that promoter recognition and strength prediction mainly rely on the accommodation of functional motifs. This work provides DL tools to design promoters and assess their functions, paving the way for DL-assisted metabolic engineering.

X. Wang, K. Xu, Y. Tan, S. Yu, X. Zhao, J. Zhou
Engineering Research Center of Ministry of Education on Food Synthetic Biotechnology and School of Biotechnology
Jiangnan University
1800 Lihu Road, Wuxi, Jiangsu 214122, China
E-mail: zhoujw1982@jiangnan.edu.cn
X. Wang, K. Xu, Y. Tan, S. Yu, X. Zhao, J. Zhou
Science Center for Future Foods
Jiangnan University
1800 Lihu Road, Wuxi, Jiangsu 214122, China
J. Zhou
Jiangsu Province Engineering Research Center of Food Synthetic Biotechnology
Jiangnan University
Wuxi 214122, China

## 1. Introduction

Promoters are DNA sequences that regulate gene transcription.[1] The activity of promoters affects both protein expression levels and secretion efficiency. Controlling gene expression levels is important for recombinant protein expression and engineering of microbial cell factories.[2–4] However, there is a limited number of naturally existing strong promoters to facilitate high protein expression. Current techniques mainly rely on mutagenesis or shuffling of key elements for screening novel promoters, which is labor-intensive. Controllable design of stronger promoters is beneficial but remains challenging. Previously, a generative deep learning (DL) model was implemented for the de novo design of highly diversified promoters,[5,6] suggesting combined DL models may provide solutions for controllable design of active promoters. However, it is still necessary to further explore the possibility of using DL models for de novo promoter design, in order to address previous limitations and increase the number of available strong promoters.

Understanding of the mechanism behind promoter activity has contributed to promoter research and design. Fundamental studies have suggested that promoter activity is mainly affected by the accommodation of functional motifs,[7] RNA polymerase (RNAP) occupancy,[8] and the distribution of sequential fragments.[9] The activity of constitutive promoters is also related to the concentration of free RNAP and their minimum requisite on RNAP concentration.[10] Several bioinformatics tools have been developed to analyze the sequential composition rules for regulating promoter activity.[9,11–15] Recognition of the critical components that affect promoter activity is pivotal for promoter design. Bacterial promoters contain key functional motifs known as −10 and −35 motifs that are recognized by RNAP to facilitate RNA transcription, which affects protein expression levels.[16,17] Therefore, in several studies promoters were engineered by shuffling the −35 and −10 motifs. Joseph et al.[7] suggested that the combination of TTTACG (−35 motif) and TATAAT (−10 motif) displayed higher promoter activity in *Escherichia coli* (*E. coli*) than the other 6-mer combinations. In addition, the spacer length between the −35 and −10 motifs can also critically affect promoter activity.[18] Excessively long and short spacers can both contribute to weak promoter activity, while 17-bp spacers are most likely

to form high-strength promoters.[18] Previous studies illustrated several subtypes of *E. coli* promoters with distinct functions,[19] such as inducing transcription at a certain growth phase or under heat shock. Therefore, accurate prediction of promoter activity across different subtypes remains challenging, despite the availability of several tools displaying high accuracy in predicting strong and weak promoters.[5,20]

Bioinformatics provides potential solutions for accurate and controllable design of promoters compared with classic methods relying on mutagenesis or engineering of functional motifs.[7,21–23] A growing number of DL methods have been developed for predicting promoter strength[20,24–26] and generating novel promoters.[5,27] However, many of the developed models were not adopted for practical engineering of active promoters. Wang et al.[5] adopted a supervised model to predict promoter transcription levels and generate promoters using a generative adversarial network (GAN). The generated promoters with predicted high activity were experimentally validated and aligned with a reported strong constitutive promoter, J23119.[28] However, many of the generated promoters displayed very low activity, suggesting the need for further exploration of a better generative model or other tools to assist in the promoter design task.[5] GAN was also implemented for the architecting of deep exploration networks (DENs) by Seelig et al.,[6] which optimized their capability for engineering desired promoter sequences. Generative models combined with supervised models provide possibilities for artificially designing promoters. However, further research is needed to determine to what extent DL methods can contribute to the generation of functional promoters.

In the present study, we aimed to develop tools for the de novo design of active promoters and for the prediction of their strength. We introduce two approaches, named "DRSAdesign" and "Ndesign" (**Figure 1**). DRSAdesign relies on a generative network and strength prediction tools for the de novo design of promoters.[5] We integrated a diffusion model-based generative network into DRSAdesign, which aims to address several limitations of previous generative models. For example, GAN is unstable during training and it is hard for variational autoencoders to generate high-quality samples.[29,30] Taking advantage of promoter prediction models,[20,24,25] we developed tools to adopt a promoter length of 50 bp, rather than 81 bp in a previous study, to fit the promoter-generating model. These tools are used to predict whether promoters are real or fake as well as their strength. Notably, DRSAdesign is based on a top-down process, in which generated promoters are classified as real or fake, followed by prediction of their strength. This process was developed because single models may not provide sophisticated results. Simultaneously, constraint-based generation of sigma70 promoters was conducted to train our developed supervised model, and the model was further used to predict the strength of novel generated promoters; this method is named Ndesign (Figure 1).

## 2. Results

### 2.1. Developing Supervised Models for Promoter Strength Prediction

PromoS was built based on Residual Network (ResNet)[31] and self-attention[32] (**Figure 2**A) for classification of promoters as strong and weak. This network was trained using RegulonDB[33] by adjusting the promoter length to 50 bp. We attempted sequential feature extraction using a one-hot encoding strategy, pseudo-dinucleotide composition (pseDNC),[25] and the combined methods (Figure S1, Supporting Information). Meanwhile, we used a convolutional neural network (CNN), BiLSTM, ResNet, and self-attention for network construction. The network was trained using a promoter length of 50 bp, and the architecture was optimized based on the tenfold cross-validation accuracy. We used the one-hot method for feature extraction, and a deep network with ResNet and self-attention integrated provided better performance with an accuracy of 0.7806 (Figure 2A and Figure S2, Supporting Information). Different model architectures trained using promoters with a length of 81 bp also suggested that integration of ResNet and self-attention improves performance based on tenfold cross-validation (Figure 2B and Figure S3, Supporting Information). Model robustness was validated by comparison with a previous state-of-the-art method trained with longer promoters.[20] We showed that PromoS trained with short promoters can still make accurate predictions (Figure 2C).

A recent study illustrated that gene expression was highly correlated with transcription in *E. coli*.[34] Another model, PromoA, was created and trained using promoter transcription level data. The dataset, which we named NDB, contains *E. coli* native promoters obtained from global TSS maps created by Thomason et al.[35] The database contains 11884 non-redundant promoters, of which 3098 are not constitutive.[35] Regarding sample diversity and complexity, PromoA achieved a Pearson correlation coefficient (PCC) of 0.31 by 10-fold validation (Figure 2D and Figure S4, Supporting Information), which was higher than the previously reported value of 0.25.[5] To test whether the downstream region of the promoter can interrupt transcription, we increased the promoter length to 81, 100, 150, 200, 300, and 400 bp by adding certain downstream regions from the *E. coli* K-12 genome.[36] We found that longer promoter sequences for training did not increase PCC (Figure 2E).

Next, model performance was analyzed on different promoter types. RegulonDB[33] and NDB[35] both contain all types of promoters. The vast majority of sigma promoters existing naturally are sigma70 promoters. Because the promoters in NDB were not classified, we adopted SAPPHIRE[37] to classify the promoters in NDB as sigma70 and non-sigma70 (Supplementary File 1). In RegulonDB, 48.5% of the promoters were sigma70 promoters, and in NDB, 37.4% of the promoters were sigma70 promoters (Figure S5, Supporting Information). The best leads from the 10-fold cross-validation of PromoS and PromoA were used. PromoS achieved an accuracy of 0.74 for sigma70 promoters and 0.81 for non-sigma70 promoters, whereas PromoA achieved a PCC of 0.314 for sigma70 promoters and 0.306 for non-sigma70 promoters (Figure S5, Supporting Information).

### 2.2. Designing a Supervised Model for Predicting if Promoters Are Real or Fake

PromoR was developed to predict if promoters are real or fake. The purpose of training PromoR is to understand the key elements of which real promoters consist and avoid designing fake but highly active promoters. Fake promoters are

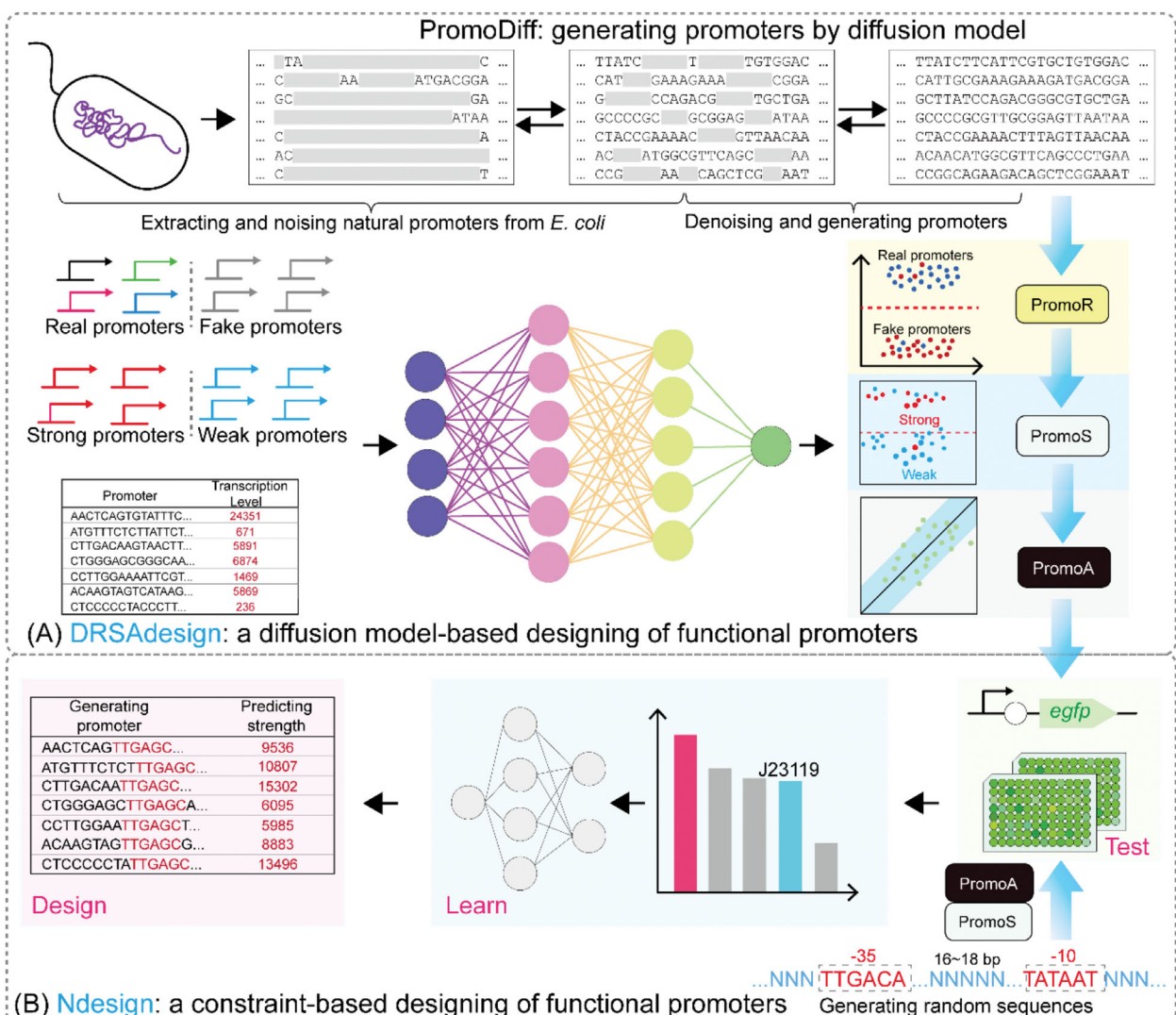

**Figure 1.** Workflow of promoter design and screening high-activity promoters. A) Promoter design was carried out by building a diffusion model to learn features from natural promoters to generate diversified novel promoters. First, it was predicted whether the generated promoters were real or fake using PromoR, followed by predicting whether they were strong or weak and transcription levels using PromoS and PromoA, respectively. The predicted real promoters with high activity were experimentally validated by inducing *egfp* expression. B) Random promoters were generated by constraining −35 and −10 motifs and the spacer length. The activity of the promoters was characterized by inducing *egfp* expression. A network was built to learn features from the constraint-based functional promoters. The network was further used for predicting the strength of novel promoters and screening high-activity promoters.

non-promoter sequences without promoter activity as annotated in RegulonDB,[33] which were obtained from the middle regions of long coding sequences and convergent intergenic regions in the *E. coli* K-12 genome.[38] PromoR was initially trained using 3382 promoters and 3382 non-promoter sequences from RegulonDB[33] by adjusting the promoter length to 50 bp. Through five-fold cross-validation, PromoR achieved an accuracy of 0.86, which is comparable to the highest reported value of 0.8603[20] (Table S1, Supporting Information). However, PromoR displayed an accuracy of 0.767 on NDB,[35] suggesting its weak performance on diversified samples. To make PromoR a robust tool, a combined dataset consisting of RegulonDB, NDB, and 11884 non-promoters was organized (totally 30532 samples). The 11884 non-promoters were obtained by randomly

retrieving 50 bp from the 81 bp of non-promoter sequences in RegulonDB.[33] PromoR trained with the combined dataset achieved an accuracy of 0.8861 using five-fold cross-validation (Table S1 and Figure S6, Supporting Information), which was obviously higher than that of PromoR trained with the smaller dataset and the previously reported value.[20]

### 2.3. Constructing an Unconditional Diffusion Model for Generating Promoters

We introduced PromoDiff, an unconditional diffusion model-based network for end-to-end generation of promoters by learning features from natural promoters. In this case, uncon-

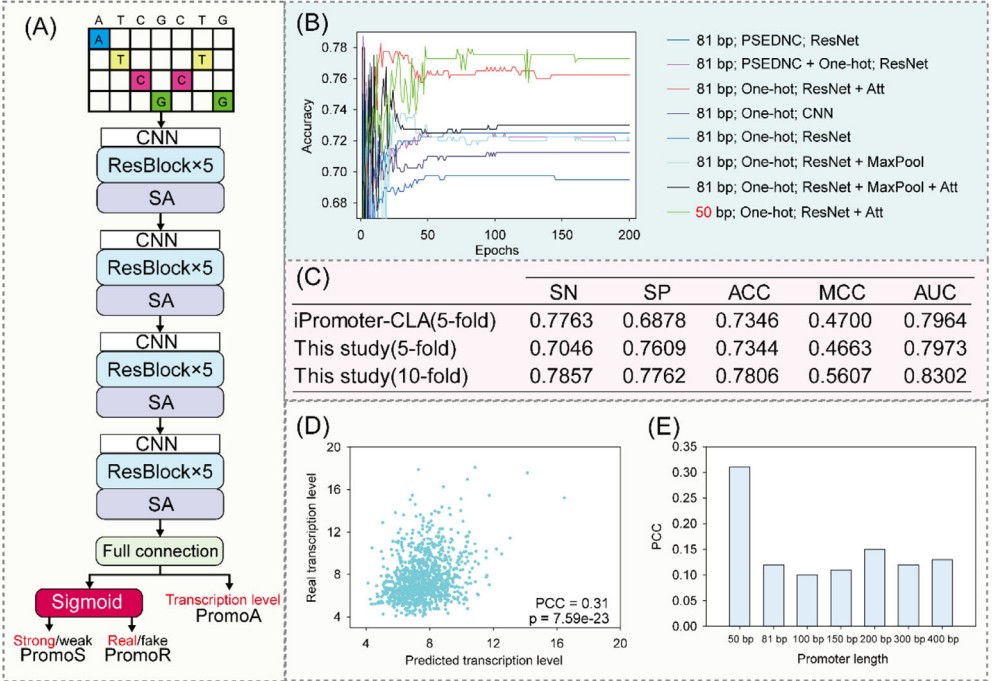

**Figure 2.** Supervised models for promoter activity prediction. A) The architecture of supervised models for promoter recognition and strength prediction. The CNN block contains a convolutional layer and a Rectified Linear Unit (ReLU) layer; the ResBlock contains two CNN layers; the self-attention block captures dependencies between different positions in a sequence. For binary classification, a sigmoid layer was added. B) The network of PromoS was optimized by attempting different feature extraction methods and network architectures, where pseDNC represents pseudo-dinucleotide composition,[25] seq represents one-hot encoding, and Att represents self-attention. C) A comparison of the ability of iPromoter-CLA and PromoS to predict whether promoters are strong or weak. Notably, iPromoter-CLA used promoters with a length of 81 bp for training while PromoS used promoters with a length of 50 bp. D) The correlation of predicted transcription levels with real levels induced by natural promoters. E) The influence of promoter length on PromoA prediction. *P* was calculated using the regression test in Excel 2013.

ditional promoter generation indicates that the model converts noise into any random representative data sample without any guide. Thus, the unconditional model can generate a promoter of any nature. A diffusion model is used on promoters to gradually add Gaussian noise to the input data point ($X_0$) by a series of steps ($Z$), and the model learned to reverse the noising process which can recover the sample to the denoised state (**Figure 3**A). The diffusion model is basically integrated by UNet for extracting features from real samples and generating samples mimicking the input ones with variations.[29] To improve the performance of the diffusion model, ResNet and self-attention were adopted for architecting the encoder side of UNet, whereas upsampling was used to recover the sample size in the decoder side.

PromoDiff was trained using NDB, and the quality of the generated promoters was evaluated in two ways: 1) PromoR predicted the real promoter portion (RPP); and 2) a sequence logo (SL)[39] was generated to visualize the conserved motifs. It should be noted that NDB contains different types of promoters (Figure S5, Supporting Information). Thus, PromoDiff as an unconditional model can generate any type of *E. coli* promoter present in the training set. PromoDiff was initially trained for 5000 epochs and the RPP values stabilized after 500 epochs (Figure 3B). Generated promoters showed less chaotic SL after 200 epochs of training (Figure 3C and Figure S7, Supporting Information). We optimized the learning rate according to RPP, which showed that

promoters generated within epoch 600 to 800 had the highest RPP (Figure S8, Supporting Information). In addition, epoch 620 showed the highest RPP and the SL was less chaotic (Figure S9, Supporting Information). We further generated 25 000 promoters adopting epoch 620, Visualization of the SL revealed that these promoters had the same −10 motif (TATAAT) as the training samples, but the −35 motif varied from TT to AT or TA (represented as "All" in **Figure 4**). Meanwhile, generated promoters with unexpected logos aligned with the training set, which showed high GC content within the −50 to −35 region and high A content within the −10 to 0 region. Notably, the generated promoters showed non-redundancy with the training set (Supplementary File 2).

Extraction of predicted real promoters (using PromoR) from the generated promoters showed that unexpected logos were significantly minimized (represented as "Real" in Figure 3C). For the predicted real promoters, their strength was predicted for experimental validation. The method named DRSAdesign, which is based on a combination of a diffusion model and supervised models for generating promoters and predicting their activity, was used. Green fluorescent protein (GFP) is widely adopted for characterizing promoter strength.[5,40] In the present study, we used enhanced GFP (EGFP) to evaluate the strength of generated promoters.[41] We selected promoters with PromoS scores above 0.9 and the top 50 PromoA scores for inducing *egfp* expression. Promoter activity was calculated as fluorescence

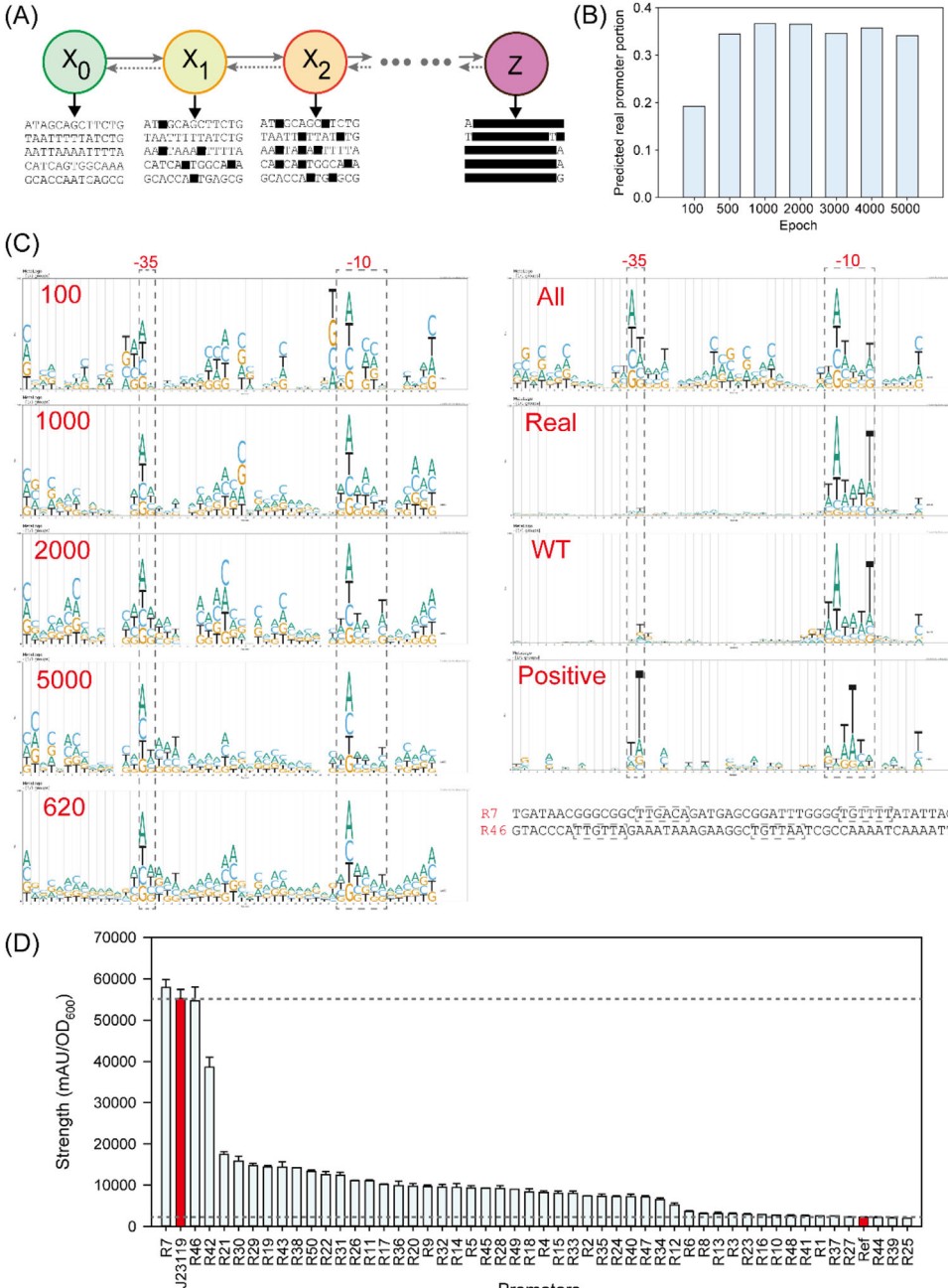

**Figure 3.** Diffusion model-based promoter design. A) A simple representation of the diffusion process. The samples in the initial state were continuously noised until the final state, and the samples were denoised to recover samples that resembled the original samples. B) The predicted real promoter portion during diffusion model training. C) The sequence logo of generated promoters during diffusion model training. The promoters generated using PromoDiff during network training were named according to their training epochs; for example, "100" indicates promoters generated by epoch 100. "All" represents all 25 000 promoters generated using PromoDiff (model deposited by epoch 620). "Real" represents all promoters predicted as real promoters by PromoR. "WT" represents all promoters from NDB.[35] "Positive" represents the validated functional promoters. D) Generated promoters were characterized by inducing *egfp* expression. The error bar shows the standard deviation of six biological replicates.

intensity (FI) divided by the optical density at 600 nm ($OD_{600}$). Five promoters showed a less than 20% increase in $FI/OD_{600}$ compared with the negative control (a non-promoter sequence) and were hence considered as non-active promoters (Figure 3D). The strong constitutive promoter J23119 was used as a positive control to evaluate if generated promoters can have high activity.

Our results indicated that 90% of the validated promoters were active, but most of them were weaker than J23119 (Figure 3D and Supplementary File 3). SL analysis indicated that the active promoters had significant logos for −35 and −10 region, which were TT and ATTTTA, respectively (Figure 3C). The strongest screened promoter, R7, displayed 5% higher activity than J23119.

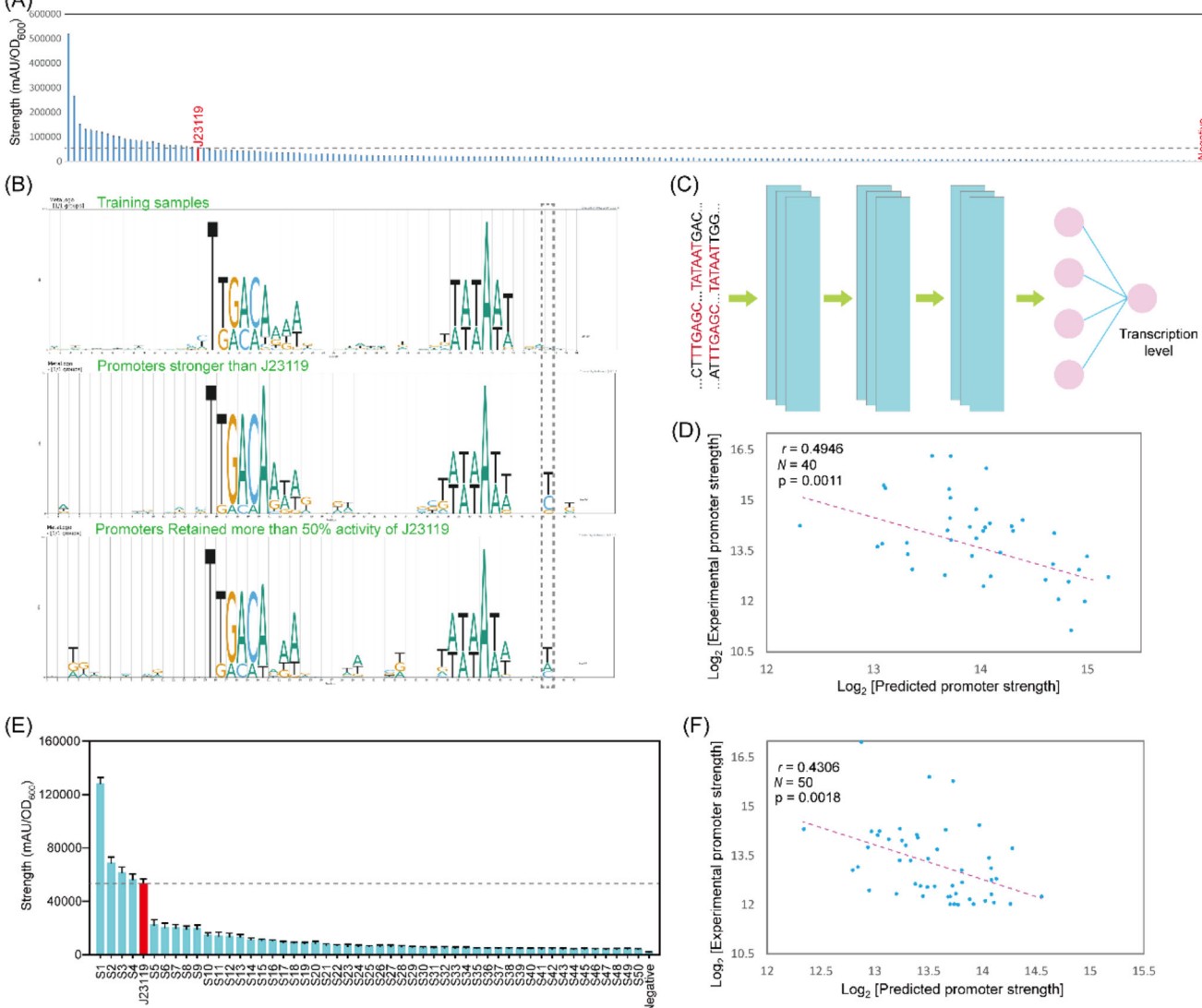

**Figure 4.** Constraint-based promoter design. A) Generating random promoters by constraining the −35 and −10 motifs to TTGACA and TATAAT with a spacer length of 16–18 bp, and experimental validation by using the generated promoters to induce *egfp* expression. The promoters are named P1 to P50 from left to right on the *x*-axis. B) The sequence logo of the generated 200 random promoters (top), the validated promoters stronger than J23119 (middle), and the promoters retaining 50–100% of J23119 activity (bottom). C) A simple CNN network for promoter activity prediction. D) The correlation of predicted and real promoter strength of the 200 randomly generated promoters. E) The activity of 50 randomly generated promoters. F) The correlation of predicted and real promoter strength of the 50 randomly generated promoters. The error bar shows the standard deviation of six biological replicates.

## 2.4. Constraint-Based Generation of Sigma70 Promoters Using PromoDiff

The weak correlation between predicted and real promoter strength has inspired our further research to simplify the design protocol by learning from data with distinctive features. *E. coli* adopts the same RNAP which can associate with different sigma factors to initiate gene transcription.[42] Among these sigma factors, sigma70 is the housekeeping sigma factor that induces transcription of essential bacterial growth-related genes.[19] The design of sigma70 promoters was previously reported,[11] and the conserved sigma70 motifs were suggested to benefit RNAP binding and transcription efficiency.[7,9,14,19] In the present study,

random promoters were generated by maintaining the conserved −35 (TTGACA) and −10 (TATAAT) motifs of the sigma70 promoter and the spacer length varied from 16 to 18 bp. To explore the sequential content, we generated 1 500 000 promoters and selected those with a PromoS score of 1 and the top 200 PromoA scores at different spacer lengths for inducing *egfp* expression.

Our results indicated that 99.1% of the randomly generated promoters were active, with more than 20% higher FI/OD$_{600}$ values than the negative control. By contrast, two promoters displayed only 17.1% and 11.5% higher FI/OD$_{600}$ values than the negative control and were hence considered as inactive promoters (Figure 4A). In total, 23 promoters (11.5%) displayed higher activity than J23119; of these, eight promoters exhibited an

increase in activity of more than 100% (Figure 4A and Supplementary File 3). Promoters P1 and P2 exhibited 8.74- and 3.96-fold higher activity than J23119, respectively (Figure 4A and Supplementary File 3). Of the promoters with higher activity than J23119, 52% had a spacer length of 16 bp, including the strongest promoter P1, while the remaining 48% had a spacer length of 17 bp. Moreover, 60% of the promoters that exhibited activity values of more than 50% of that of J23119 also had a spacer length of 16 bp, suggesting 16 bp may be an optimal spacer length according to the results in this study. No obvious logos except for functional motifs were found for all training promoters, and a conserved T was found after the −10 motif (Figure 4B).

### 2.5. Learning- and Constraint-Based Design of High-Activity Promoters

To generate a model that can accurately predict promoter activity, a simple network named PromoNet was built by only integrating CNN (Figure 4C). PromoNet learned features from the 200 constraint-based randomly generated promoters, which displays a PCC of 0.4946 using five-fold cross-validation (Figure 4D and Figure S10, Supporting Information). To validate the accuracy of the network, the strength of 50 constraint-based random promoters was predicted using PromoNet. Experimental validation showed four promoters with higher activity than J23119, with promoter S1 showing a 43% increase in activity (Figure 4E). The other generated promoters exhibited activity of less than 50% of that of J23119. The PCC of prediction and experimental validation was 0.4306 (Figure 4F), suggesting that the network trained using a small dataset can also learn features for prediction. The method for constraint-based generation of random promoters and prediction of their activity is named "Ndesign."

### 2.6. Identification of Random 6-mers as Potential Functional Motif for Sigma70 Promoters

Up to 99.1% of constraint-based generated random promoters were active, highlighting the significance of functional motifs. We adopted PromoR and PromoS to further predict potential 6-mers that may be useful for constructing sigma70 promoters. First, we maintained the conserved −35 motif (TTGACA) to investigate possible 6-mers that can comprise the −10 motif. Random promoters were generated using an optimal spacer length of 17 bp according to the previous study[43] (**Figure 5**A). Of all possible 6-mers combined with the conserved −35 motif, 6.91% have a 100% chance of forming real promoters (Figure 5B and Supplementary File 4). Meanwhile, 66.2% of all possible 6-mers have a >50% chance of forming real promoters, and these 6-mers have a 71.66% chance of forming strong promoters (Figure 5B). In comparison, the 6-mers with a <50% chance of forming real promoters still have a 68.8% chance of forming strong promoters. A previous study highlighted the contribution of −35 and −10 motifs to promoter function by investigating their binding to RNAP/sigma70.[27] We showed that 70% of the −10 motifs with the strongest binding to RNAP/sigma70 have a 95% chance of forming real promoters, whereas 55% of the −10 motifs with the weakest binding to RNAP/sigma70 have a <50% chance of forming real promoters (Figure 5C). The −10 motifs with the strongest binding showed slightly higher chances of forming strong promoters (71.65%) than the −10 motifs with the weakest binding (66.75%).

Next, the role of the −35 motif was tested by maintaining the conserved −10 motif (TATAAT). This time, all generated random promoters (including random 6-mers) with TATAAT at the −35 position were predicted as real promoters, and only 8.54% of TATAAT in combination with other possible 6-mers were predicted as weak promoters (Supplementary File 4). However, PromoR predicted that RPP decreased obviously when changing TATAAT to other 6-mers.

## 3. Discussion

In the present study, DL models were generated for *E. coli* promoter design. Previously developed DL tools showed high accuracy in promoter recognition and strength prediction.[20,24,44–45] However, experimental evaluation of these tools was still lacking. In the present study, we introduced DRSAdesign to validate the capability of DL tools to design promoters. DRSAdesign was built by integrating generative network and supervised models inspired by previous studies.[5,6] We showed the integrated models had high accuracy compared with previously reported models (Figure 2 and Table S1, Supporting Information), indicating that adopting a promoter length of 50 bp for generating novel promoters and predicting their properties is feasible. To address the limitations, including the positive rate of the generated promoters, the accuracy of supervised models, and stabilization of the generative network, we attempted to predict whether promoters are real or fake, optimized the activity prediction model, and used a diffusion model (which is suggested to be more stable than GAN[29,30]) for generating novel promoters in this study. Even though 90% of the generated promoters were active, their activity varied by more than 150-fold; regardless, they were all predicted as strong promoters (Figure 3D and Figure 4A), suggesting the weak performance of promoter strength prediction tools despite their high in silico accuracy (Figure 2 and Figure 3).

On the one side, the training data for both PromoA and PromoS were *E. coli* natural promoters with single copies or low copy numbers in the genome. Promoter activity can be changed dramatically when the copy number increases. On the other side, splitting the promoters in RegulonDB and NDB showed that constitutive promoters only took a portion of less than half in the given dataset, while the other promoters contained other sigma types or unknown active promoters (Figure S4, Supporting Information), indicating a lack of sophisticated DL models to capture features from the variety of dissimilar data. Moreover, the portion of sigma70 promoters in NDB is significantly less than that in RegulonDB, and regression prediction is harder than binary prediction, which together contributed to the low accuracy of PromoA. To clarify if the DL model was properly used, the original data for PromoA and PromoNet were used to train machine learning models including elastic net regression and gradient boosted trees. Our results showed that the achieved PCC for each machine learning model was below 0.1 after

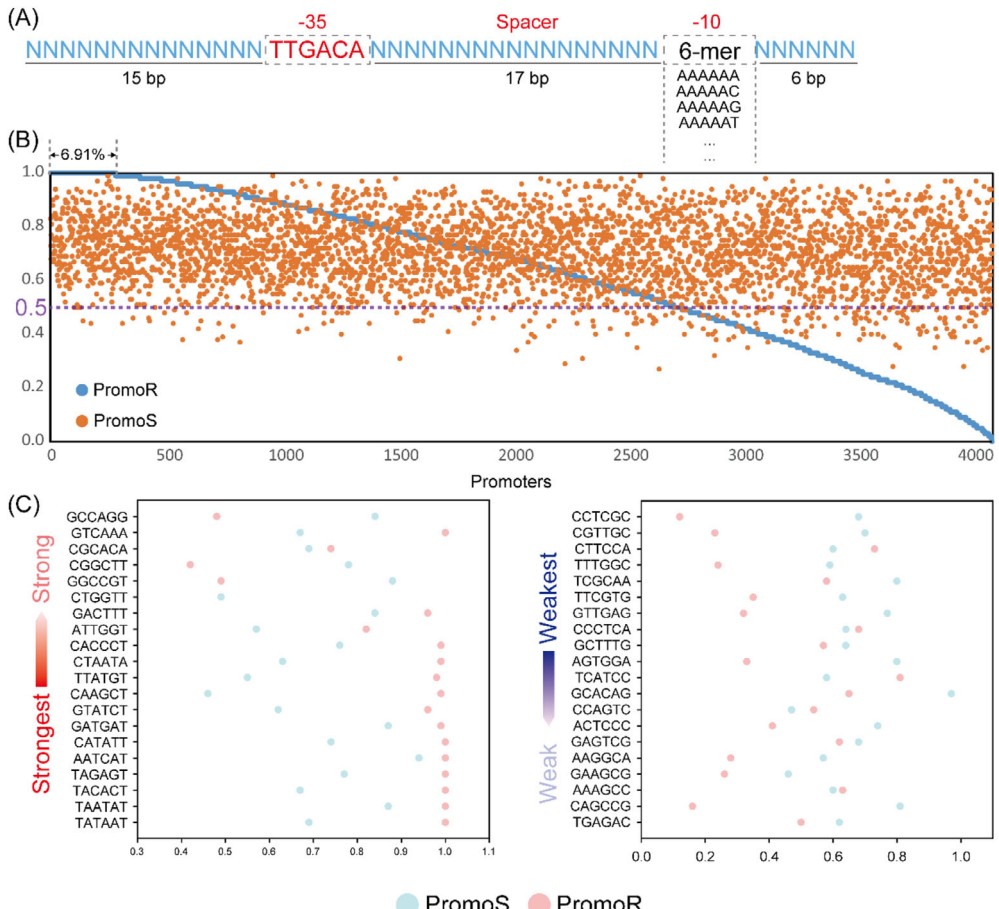

**Figure 5.** Analysis of possible 6-mers as sigma70 functional motifs. A) Generating random promoters by constraining −35 and −10 motifs and the spacer length. B) Random promoter recognition and strength prediction using PromoR and PromoS, respectively. C) The 20−10 motifs showing the strongest and weakest binding from a previous study:[27] the portion of predicted real promoters and the portion of predicted strong promoters of the randomly generated promoters by constraining −35 and −10 motifs.

hyperparameter optimization (Table S2, Supporting Information), which is much lower than that of PromoA and PromoNet.

Additionally, we confirmed the importance of data composition and sample size for model performance.[46] First, several models trained with RegulonDB showed minor accuracy differences, varying from 0.7073 to 0.7346.[20,24,47,48] Second, generative models like the GAN-based model[5] or PromoDiff trained using NDB can generate highly diversified promoters, but most of these promoters had low activity, like the original training data. Third, PromoR trained with a large and diversified dataset displayed better performance than PromoR trained with the original small training set.[49]

Based on PromoR prediction, 23.5% of the random sequences carrying all possible 6-mers as the −10 motif and the conserved −35 motif (TTGACA) had a 90% chance to form real promoters (Supplementary File 4). In addition, 100% of the random sequences carrying all possible 6-mers as the −35 position and the conserved −10 motif (TATAAT) can form real promoters (Supplementary File 4). These results indicate that PromoR recognizes certain 6-mers as a critical standard to evaluate if the given sequences can be real promoters. This may be disadvantageous for prediction because non-promoters can contain 6-mers like

TATAAT. Moreover, the 6-mers that easily formed active promoters did not have a high chance to form strong promoters, indicating that the sequence composition other than the functional motifs is also important for promoter strength prediction.[43]

The models trained with NDB showed less correlation between predicted and real promoter activity,[5] which inspired us to organize data with distinctive features for model training. Sigma70 promoters have characteristic sequence elements and account for the major portion in the training sets used in this study. Moreover, PromoS and PromoA displayed high accuracy against sigma70 promoters. Therefore, we considered to generate random promoters carrying motifs of sigma70 promoters to train the PromoS- and PromoA-derived network PromoNet. For the regression prediction, PromoNet achieved a PCC of 0.4946, which is higher than that of the model trained using NDB (0.31). Intriguingly, all of the generated strong promoters had a spacer length of 16 or 17 bp, indicating that: 1) 17 bp is not the only proper spacer length and 2) a length of 18 bp is not favorable for strong promoters.[9,18] Many natural *E. coli* promoters have a spacer length of 18 bp, which may serve as an evolutionary strategy to control downstream gene expression.[50] Even though PromoNet trained with constraint-based promoters gave a higher

PCC, the low amount of training data may limit the performance of PromoNet for the exploration of other sequences than the −35 and −10 motifs.

## 4. Conclusion

Our study shows that DL tools can be implemented for characterizing promoter functions and guiding promoter design. DL tools are more robust for promoter recognition than promoter strength prediction. We highlight the importance of data regularity for model accuracy. The DL tools designed in this study can assist in the design of novel promoters and provide guidance for the development of other networks.

## 5. Experimental Section

*Promoter Library Construction and Recombinant Protein Expression*: *E. coli* strain BL21 was used as a recombinant protein expression platform. *egfp*[41] was synthesized and cloned into the *NcoI* and *XhoI* sites (Figure S10A, Supporting Information) of pET-28a (Genscript, Nanjing, China). Linear DNA with homologous arms was amplified by PCR using pET-28a/*egfp* as a template. Promoter DNA was synthesized as single-stranded complementary primers (Table S3, Supporting Information) and annealed using PCR. Promoter DNA was ligated into the linear vector using one-step cloning (Figure S10B, Supporting Information). The Direct PCR Kit (Sangon, Shanghai, China) was used for PCR following the manufacturer's instructions. Circularized plasmids were chemically transformed into *E. coli*. Initial cultivation was carried out at 37 °C for 12 h in Luria–Bertani medium containing 100 µg mL$^{-1}$ of kanamycin, followed by transfer to Terrific-broth medium and cultivation at 30 °C for 10 h. Cells were cultivated in 24-well plates, and six replicates of each variant were cultivated for validation. A non-promoter sequence was used as the negative control, and the J23119 promoter was used as the positive control.

*Determination of Promoter Activity*: Cell density was determined by measuring OD$_{600}$ using a microplate reader (BIOTEK, Cytation 3). The cells were collected by centrifuging fermentation samples at 12 000 *g* for 5 min and resuspended with Tris-HCl (pH 8.0). The cells were sonicated using ultrasonic oscillation (Sonics VCX750, amplitude 38%). The obtained cell lysate was centrifuged at 12 000 *g* for 5 min and the supernatant was collected to determine the intracellular eGFP levels. FI was measured at an excitation wavelength of 395 nm and an emission wavelength of 509 nm using a microplate reader (BIOTEK, Cytation 3). Promoter activity was calculated as FI divided by OD$_{600}$.

$$\text{Promoter activity} = \frac{FI}{OD_{600}} \tag{1}$$

*Dataset*: The strong and weak promoter data were obtained from RegulonDB, which contains 1591 strong and 1792 weak promoters.[33] This dataset also contains 3382 fake promoter sequences. The promoters in RegulonDB were originally 81 bp in length, but they were reduced to 50 bp according to their −35 and −10 motifs distribution (Figure S12, Supporting Information). The fake promoter sequences were randomly retrieved to obtain 15 266 fake promoters with a length of 50 bp.

NDB contains 11 884 non-redundant samples which were 50 bp upstream of the TSS.[5,35] Note that 6297 of the promoters were constitutive promoters, while the others are active during certain cell phases. The prediction of −35 and −10 elements was conducted using SAPPHIRE,[37] which showed that only 4442 promoters had standard sigma70 −35 and −10 motifs (Supplementary File 1). A majority of the natural promoters (97.7%) had a transcription activity of <10 000 (Figure S13, Supporting Information). To investigate whether the length of the downstream promoter sequence affected activity, the length of the promoters was extended to 81, 100, 150, 200, 300, and 400 bp by extracting sequences from the *E. coli* K-12 genome.[36]

*Promoter Feature Extraction*: Sequential feature extraction was carried out using one-hot encoding and the pseDNC[25] method (Figure S1, Supporting Information). One-hot encoding refers to using a vector with the length of the categories in the dataset; if a data point belongs to one of the categories it is classified as 1; otherwise, it is classified as 0 (Figure S1A, Supporting Information). PseDNC uses a value to describe the physical structure of two neighboring nucleotides; the value can be affected by the position of the two nucleotides in the sequence and the type of steric interaction of the two nucleotides such as roll, slide, and shift (Figure S1B,C, Supporting Information).

*Network Architecture*: The networks for strong and weak promoter classification (PromoS), transcription level prediction (PromoA), and promoter recognition (PromoR) were all built by integrating ResNet[31] and self-attention.[32] Hyperparameter optimization was carried out by optimizing the learning rate and batch size after the model was developed (Figure S2, Supporting Information). In short, learning rate and batch size were optimized from 0.0001 to 1 and from 4 to 64, respectively. ResNet is a deep CNN which was developed to cope with the problem of vanishing gradient by introducing a skip connection and summing subsequent layers. In the present study, ResNet was implemented by building 20 ResBlocks between the CNN block and the self-attention block (Supplementary File 5). The formula of ResNet is given below, where *x* represents the input, *F(x)* represents the output from the layer, *Wi* represents the parameters given to the CNN layer, and *Ws* represents certain convolution configurations to make the dimensions of input and output identical.

$$Y = F(x, (Wi)) + Ws \tag{2}$$

The self-attention mechanism is used to link the significance of the partial information to the overall information. The equation is given below, where *Q*, *K*, and *V* are vectors of queries, keys, and values of dimension $d_k$, where $d_k$ is the size of the attention keys.

$$\text{Attention}(Q, K, V) = \text{softmax}\left(\frac{QKT}{\sqrt{dk}}\right)V \tag{3}$$

Due to all three networks being shared with the same sample length and type, initial network optimization was carried out based on PromoS, and the other two networks adopted the same architecture with minor differences. For network optimization, the prediction accuracy of PromoS was used to provide feedback. CNN, ResNet, self-attention, MaxPool, and dropout modules were integrated in the model. Promoter features extracted using one-hot encoding and pseDNC were independently or combined used as input, and an output value of 1 was considered as a strong promoter and an output value of 0 was considered as a weak promoter. The mean squared error (MSE) was used for optimization.

PromoDiff was a diffusion model with UNet integrated. UNet was created for semantic segmentation. It contained an encoder and decoder block for feature extraction and sample size recovery. To learn features of promoters, ResNet and self-attention were used on the encoder side, and to recover the promoter size we adopted upsampling on the decoder side. For the diffusion model,[29] the features of an input promoter were extracted using the one-hot encoding method. The data point was denoted as $X_0$. The forward diffusion process was based on gradually adding Gaussian noise to $X_0$ through *T* steps, and the Gaussian noise with variance can produce a new latent variable $X_t$. Then the developed network was trained to recover the original data by reversing the noising process. The reverse diffusion process can generate new data which highly mimic the original ones.

*Model Evaluation*: The accuracies of PromoS and PromoR for strong and weak promoter prediction and real and fake classification were calculated by dividing the number of correct predictions by the total number of samples. The accuracy of PromoA was the PCC of predicted and real promoter activity. The performance of PromoS was evaluated using sensitivity (Sn), specificity (Sp), accuracy (Acc), Mathew's correlation

coefficient (MCC), and the receiver operating characteristic (ROC) curve as previously reported.[20]

$$Sn = \frac{TP}{TP + FN} \tag{4}$$

$$Sp = \frac{TN}{TN + FP} \tag{5}$$

$$Acc = \frac{TP + TN}{TP + TN + FP + FN} \tag{6}$$

$$MCC = \frac{TP \times TN - FP \times FN}{\sqrt{(TP + FP) \times (TP + FN) \times (TN + FP) \times (TN + FN)}} \tag{7}$$

## Supporting Information

Supporting Information is available from the Wiley Online Library or from the author.

## Acknowledgements

This study was funded by the National Key Research and Development Program of China (2019YFA0904900) and the Starry Night Science Fund of Zhejiang University Shanghai Institute for Advanced Study (Grant No. SN-ZJU-SIAS-0013).

## Conflict of Interest

The authors declare no conflict of interest.

## Author Contributions

X.W. and J.Z. conceived the project and wrote the manuscript. X.W., K.X.,Y.T., S.Y., and X.Z. designed and performed all the experiments. X.W. and J.Z. analyzed the results.

## Data Availability Statement

The authors declare that all data supporting the findings of this study are available in the article and its supplementary files or are available from the corresponding author on request. The sigma promoter distribution in RegulonDB, the dataset created by Thomason et al., and the SAPPHIRE-predicted promoter types are provided in Supplementary File 1. The promoters generated using the diffusion model and promoters created by Thomason et al. are provided in Supplementary File 2. Promoter activity data are provided in Supplementary File 3. All possible 6-mers in combination with conserved −35 and −10 motifs for promoter recognition and strength prediction are provided in Supplementary File 4. All detailed information for model architectures is provided in Supplementary File 5. All code and data used in this study can be found in the GitHub repository: https://github.com/wangxinglong1990/Promoter_design.

## Peer Review

The peer review history for this article is available in the Supporting Information for this article.

## Keywords

deep learning, diffusion models, promoter design, promoter recognition, promoter strength

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
