## [**Supplementary Information**: Record of Transparent Peer Review · Advanced Genetics]

Record of Transparent Peer Review

Deep learning-assisted design of novel promoters in Escherichia coli

Xinglong Wang, Kangjie Xu, Yameng Tan, Shangyang Yu, Xinyi Zhao, Jingwen Zhou*

*Corresponding author

Review timeline:

Data submitted: 12-Jun-2023

1st Editorial Decision: 30-Jun-2023

Revision Received: 29-Jul-2023

2nd Editorial Decision: 14-Aug-2023

Revision Received: 24-Aug-2023

3rd Editorial Decision: 07-Sep-2023

Revision Received: 21-Sep-2023

4th Editorial Decision: 08-Oct-2023

Revision Received: 09-Oct-2023

Accepted: 13-Oct-2023

Editor: Yuming Hu

1 st Peer Review	12-Jun-2023 to 30-Jun-2023
----------------------------

Reviewer #1: This manuscript is about generating novel promoters in Escherichia coli. The authors developed a deep learning pipeline, DRSAdesign, to generate novel E. coli promoters. This pipeline took advantage of the diffusion model, which is a class of probabilistic generative models. The authors experimentally validated 50 DRSAdesign synthesized promoters and noted that 90% of them were active with high diversity. Furthermore, the authors introduced Ndesign, a computational pipeline or a filtering/thresholding step, employed to generate novel promoters while considering constraints related to motif composition and transcriptional activity.

This paper is interesting and there are merits in the study design and results. However, I have major concerns on various issues listed below that still need to be addressed before it can be considered suitable for publication.

1. The authors should explain why de novo design of promoters is an important topic. The factors influencing promoter activity do not seem to include mRNA decay and self-cleavage.

These two factors are more closely associated with the steady state concentration of transcripts rather than the generation of transcripts (i.e. transcription activity).

2. Although the -35 and -10 motifs are important elements in many E. coli promoters, there are other important motifs in non-sigma70 promoters. For example, the motifs for sigma54 could be found in DOI: 10.1093/nar/gky010.

If non-sigma70 promoters were included in the training, validation, and testing data, then the number of promoters for each sigma factor should be clearly indicated, and the performance metrics for each sigma factor class should be provided separately.

3. The literature review is insufficient. In terms of promoter classification, there are quite a few DL methods which could achieve better performance than the ones cited by the authors.

In addition, there are other DL methods that focus on generating novel promoters. Here is an example: doi: 10.1016/j.cels.2020.05.007. Please provide a description of how this study differs from previous similar studies, as well as the unique methods employed, which contribute to the novelty of the results and findings. Could it be possible to conduct a benchmark to demonstrate the merits of this study compared with other available approaches?

4. DRSAdesign is mentioned in Abstract, Results, and Discussion, but the term could not be found in the figures of the main text. It is also confusing how DRSAdesign and Ndesign are different, and how the whole pipeline is put together, from model training to novel promoter generation. An overview diagram might help the audience to quickly grasp the whole idea of this study.
5. No clear descriptions about the rationale and functionality of the "unconditional diffusion model", or "PromoDiff", could be found. There could be other choices of DL models for creating a generative method, such as GAN, autoencoder, etc. Please explain.

Besides, in-depth statements about the design, and the working process of this diffusion model must be given, and such as the flow of x_0 to z .

It is possible that the diffusion model outputs real/known promoters (i.e. identical to training promoter sequences). It is necessary to explicitly address how such replicates are removed, the proportion of novel promoters in all generated promoters, and the extent of how the novel promoters are different in sequence contents from the real/known promoters.

6. Please explain what J23119 is.

7. In "Experimental section", please justify the usage of "egfp" as a sufficient validation platform for assessing promoter strength. What is egfp? Is this term already used by the cited reference (doi: 10.1002/iub.1041)?

8. In "Dataset". Please explain what are fake promoters. Why they are fake? What is the purpose of using fake promoters in the training?

9. In "Dataset". It appears that the authors acquired > 10,000 promoters from NDB, on which I cannot find the resource name from the cited references. Besides, the authors must explain explicitly how to confirm that all promoters contain -35 and -10 motifs. Based on the reference of the plausible source of NDB (doi:10.1128/jb.02096-14), the statements about the promoters include "The majority of pTSS, iTSS, and asTSS are preceded by $\sigma 70$ promoter elements", and "The enrichment for two T residues comprising a potential $\sigma 70$ -35 element was significantly less than what was observed for the -10 element".

10. In "Network architecture". PromoA is trained to perform a transcription-level prediction, and PromoR is trained to classify promoters. Both models were built by including an architecture consisting of ResNet and Attention layers. However, it is suspicious if PromoR can really achieve the accuracy level required for selecting novel synthetic promoters, and a thorough benchmark to compare the it with other tools such as CNNProm, IPromoter-2L, etc. might further elucidate the value of these two new models.

In addition, it is confusing why one attention layer is placed after each set of ResBlocks in PromoA/S/R. Attention layer is different from self-attention layer in that the former is to learn the dependence between each input token and the output classification. One attention might be sufficient for the classification task.

PromoR and PromoS share the same architecture at the front-end. Subsequently, they individually classify input sequences as either Strong or Weak, and either Real or Fake. Since the front-end models are shared, for each input they give respective outputs. Thus, there is a question about how to resolve this case: Fake promoter is Strong or Weak.

11. The performance robustness/variation of PromoA/S/R in the 5-fold cross-validation must be illustrated.

12. The PCC of predicted and real transcription levels is only 0.31. And a statistical test should be performed to give the p-value.

13. It is not true that "The results highly agree with PromoR predictions that 99.1% of the randomly generated promoters were active". This high precision is actually conditioned on that the novel promoters must consist of -35 TTGAGC motif and -10 TATAAT motif. What will happen if this constraint is not applied? This filtering has therefore vetoed the alternative possibility that there might be novel active promoters consisting -35 and -10 motifs carrying more sequence variations.

14. The authors used numbering in the reference list. Each serial ID should presents only one reference paper. For example, there should be just one [12], but not [12a], [12b], etc.

15. Intensive editing is required for improving the English writing of this manuscript in order to enhance readability. This suggestion includes rephrasing and restructuring sentences to ensure clear and accurate expression of meaning.

Reviewer #2: In this work the authors develop a deep learning framework to analyze the promoter properties of *E. coli*, using the model to generate in silico predictions of novel promoters. They tested a subset of the designed promoters experimentally with a fluorescent reporter assay. The accuracy of the model is not particularly impressive from what I can tell, but given the diversity of models constructed and comprehensive scope of the work, it seems like this work has substantial merit. I have a number of suggestions that I hope can be addressed to improve the manuscript.

Major Comments

- The writing needs to be improved throughout - currently many grammatical errors.
- Cross-validation or holdout datasets should be used throughout. Cross-validation is mentioned in some parts but not in others, and it's not clear when the statistics reported are just from a training vs a test dataset in all cases. Train and test performance should be reported throughout to assess overfitting.
- The novelty over previous methods and relative performance seems a potential issue that deserves clarification. From Figure 2C, it appears that their method performs very similarly but slightly worse than the previously published iPromoter-CLA? What then is the benefit of building a new model here? Is it only to then adapt the model for design with the diffusion model later?
- Performance of the model in 2D, 5D, and 5F is actually quite disappointing. For reference, can the authors repeat their training/test analysis on simpler modeling structures, such as elastic net regression and gradient boosted trees, to determine whether the scale of the data here is sufficient to properly train the deep learning models that the authors are trying to build?
- How were fake promoters generated? Were they randomly grabbed from the *E. coli* genome? This does not seem to be a very difficult learning task then, as the average location in the *E. coli* genome is not expected to have anything like a -10 or -35 box with appropriate spacer. It would be more impressive to pick genome locations that have highly similar sequences to promoters (e.g. -10-like sequences), but do not have measured promoter activity, and see if the model can learn to discern real from fake in these cases.
- Was there any kind of verification performed to make sure that their fluorescent assay is working properly? I would normally expect error bars from multiple replicates and some kind of control titration to be done at the outset.
- There were very few details on how the model was trained. Was there any hyperparameter optimization performed?

Minor Comments

- Figure 1 doesn't seem to capture how noise is added to the sequence - the 'blurring' shown in the figure is presumably not how it works, and a more direct representation would be preferred.
- 2D and 2E seem to have their captions switched.
- Why can Figure 3D not be presented as scatterplots? It is difficult to see if the trends follow expectations from the bar plots.
- I am not clear on why certain datasets were used for certain models. Some models use just RegulonDB, some use NBD as well, etc. Why would the maximum amount of available data not be used in every model?
- The authors state that one innovation in this work is "characterizing key promoter elements" (from the abstract). Where was this discussed? I was expecting a more extensive analysis of sequence determinants of promoter activity but most discussion was just related to the well-established -10 box, -35 box, and spacer, and any new knowledge discovered in this work is unclear to me.
- Line 72 - the authors cite previous work generating novel promoters with deep learning - but I thought that was the innovation of this work over previous work according to their abstract. Can the authors please clarify their innovation?
- The phrase "unconditional generating promoters" (line 77) is completely unclear to me, and resurfaces later as well. What does "unconditional" mean in this context?
- Line 157: What is UNet? This is the only place in the paper where I see it mentioned.
- Line 76: The authors begin using promoter J23119 without discussing what it is or why it's useful as a reference.
- I do not understand the point on line 265 at all. What is "high regularity data"? Is the high performance mentioned not just the result of training a model to memorize a specific dataset? If $R = 0.94$ was possible already, it would completely invalidate the need for the submitted work.

Reviewer #3: "In this study, Zhou et al. developed deep learning pipelines for generating, recognizing, and characterizing promoter strengths. They also experimentally validated the performance of their model, and identified several stronger promoters compared to the positive control, making it an intriguing and useful study. However, I have several comments that could potentially enhance the manuscript's quality.

Major:

1. Abstract: In the abstract, the authors claim that deep learning tools for promoter design and characterization still lack research. However, they cite several previously published models for promoter design and characterization. It would be helpful to discuss the differences and explain the uniqueness of this paper. Additionally, in the Discussion section, the authors mention that GAN or diffusion models mostly generate low-activity promoters based on their training dataset, while their deep learning model, PromoS, with over 20,000 training data points, cannot achieve higher activity promoters. They further validated that a simple CNN model with constraint-based random promoter generation, PromoNet, is able to accomplish this. Therefore, it is important to justify the claim of lacking deep learning models and clarify the specific benefits and advancements presented in this paper.

2. Line 96: Please mention the reason for choosing a length of 50 bp for PromoS training.
3. Please explain and discuss why multiple models (PromoR/S/A) were developed instead of a single model that can accomplish all these tasks.
4. Line 125: Please provide additional explanation about why the weak performance on NDB leads to the conclusion of weak performance on diversified samples. Furthermore, please clarify why the NDB database is considered more diversified compared to RegulonDB.
5. Line 126: What is the source of the 11,884 non-promoters mentioned?
6. Since the training dataset is derived from E. coli, is this a deep learning tool specifically designed for E. coli promoters? If so, please mention this in the abstract and the main text to avoid confusion.
7. What are the criteria used to define a strong promoter compared to a weak promoter in the training dataset of PromoS?

Minor:

1. Line 43-44: Consider revising the sentence "It is important for engineering strong promoters for the overexpression of recombinant proteins or optimization of metabolic engineering in microorganisms" or explain the necessity of engineering if the promoter is already strong.
2. Line 180: "The active promoters with -35 and -10 motifs of TT and ATTTTA (Figure 4C)" is not a complete sentence. Please revise it accordingly.
3. The author alternately uses "strength" and "activity" to refer to the ability of the promoter. It would be beneficial to maintain consistency in the terminology used.
4. Figure 4D: Please label the significance in the figure and consider adding a reference line or using different colors to differentiate between promoters with increased activity compared to the control and those without."

Reviewer #4: The authors systematically studied the features of E. coli promoters, and architected diffusion model based deep learning model for generating new promoters. Meanwhile, the authors built deep learning models for predicting the strength of promoters, which assisted the characterization of promoters, and selecting of high activity promoters. This may be the first time for attempting using diffusion model for generating promoters, which is an interesting work. Moreover, this study tested the model accuracy regarding the architected deep learning models were previously achieved high in silico accuracy. There are issues with the readability and clarity of the results, which should be addressed, we recommend to accept the paper after revising these minor issues.

- 1: Line 95, "This network was trained using RegulonDB[19] by adjusting the promoter length to 50 bp". This may be confused to the authors of why adjusting the promoter length to 50 bp.
- 2: Line 105, "Recent study illustrated gene expression was highly correlated with transcription in E. coli regardless of other factors". This is not accurate, the other factors showed minor influences to the trade-off of transcription and expression rather than "regardless".
- 3: Line 112, "We further validated the downstream DNA's interruption to transcription". This sentence is confused.

4: Line 131, "The -35 motif is conserved which mostly starts with TT followed by less conserved GAGC in constitutive promoters". A reference is needed for this.

5: Line 118, "Identification of functional motifs for promoter architecture". This section mainly describe the possibilities of using random 6-mers as -35 and -10 motif to generate active promoters, while the spacer between -35 and -10 motif is 17 bp, therefore, this is a constraint-based generating promoters. The authors should reformatting this section to avoid confusions.

6: Figure 4D, this figure can reproduced according to promoter activity as Figure 5A and Figure 5E

7: The word "de novo" should be either with or without italics, please check the whole manuscript.

8: Line 225, "but promoter strength prediction was hard to achieve especially we adopted high copy number plasmids". This is confused.

9: Line 245, "Based on PromoR prediction, 23.5% of all possible 6-mers in the -10 position assembled with -35 motif (TTGAGC) have more than 90% chance to form active promoters, while 100% of all possible 6-mers in the -35 position assembled with -10 motif (TATAAT) can form active promoters (Supplementary File 1)". This sentence is confused.

10: Line 341, the formula is clashed.

11: Figure 2C, this table should be better formatted, the values and the headings should be centered.

12: The figure legend of figure 5, egfp is underlined.

Reviewer #5: In the reviewed manuscript, authors developed supervised deep learning(DL) models by optimizing previous models for promoter recognition and strength prediction. These models were used for evaluating the key elements for architecting active promoters and investigating factors that affect promoter strength. For promoter design, a diffusion model was adopted for generating de novo promoters followed by predicting promoter activity.

The novel and original results of this work demonstrated that:

1. This study showed DL tools can be implemented for characterizing promoter functions and guiding promoter design.
2. DL tools are more robust for promoter recognition than promoter strength prediction.
3. The DL tools architected can assist the design of novel promoters and provide guidance for the other related networks.

The title of the manuscript does reflect its content. Abstract is brief enough, indicates the purpose of the work, and includes brief presentation of experiments and main results. Introduction gives some important information concerned to the problem of models for promoter recognition and strength prediction based on relevant literature and clearly presents rationale and the goal of the study.

The methods applied in the research part of the manuscript are appropriate and presents enough details allowing repeating of the experiments. Coverage of the subject is complete and well organized. There are relationships to the earlier research given in details. The figures included in the manuscript are justified.

In general, the manuscript presents original and valuable approach to the models for promoter recognition and strength prediction. The authors present good practical experience in promoters design, recognition and strength prediction and provide guidance for the other related networks.

In conclusion, I would recommend the paper for publication in the Advanced Genetics.

1st Editorial Decision 30-Jun-2023

Editorial Decision: Major revision

Recommendation of the reviewers

Reviewer #1 Recommends Major Revision

Reviewer #2 Recommends Major Revision

Reviewer #3 Recommends Minor Revision

Reviewer #4 Recommends Minor Revision

Reviewer #5 Recommends Publish without Revision

Authors' response to 1st Peer Review

29-Jul-2023

Reviewer #1:

This manuscript is about generating novel promoters in Escherichia coli. The authors developed a deep learning pipeline, DRSAdesign, to generate novel E. coli promoters. This pipeline took advantage of the diffusion model, which is a class of probabilistic generative models. The authors experimentally validated 50 DRSAdesign synthesized promoters and noted that 90% of them were active with high diversity. Furthermore, the authors introduced Ndesign, a computational pipeline or a filtering/thresholding step, employed to generate novel promoters while considering constraints related to motif composition and transcriptional activity.

This paper is interesting and there are merits in the study design and results. However, I have major concerns on various issues listed below that still need to be addressed before it can be considered suitable for publication.

Response: Thanks for your positive comment. We have carefully read all the comments, and the manuscript has been revised as suggested.

1.

The authors should explain why de novo design of promoters is an important topic. The factors influencing promoter activity do not seem to include mRNA decay and selfcleavage. These two factors are more closely associated with the steady state concentration of transcripts rather than the generation of transcripts (i.e. transcription activity).

Response: Thanks for your suggestion.

(1) The explanation for the importance of de novo design promoters was added to the first paragraph of Introduction section, as follows:

Controlling gene expression level is important for protein recombinant expression and engineering microbial cell factories.[2-4] However, there is limited number of naturally existing strong promoters to facilitate high level protein expression. Current techniques mainly relied on mutagenesis or shuffling key elements for screening novel promoters, which is labor-intensive. Controllable designing stronger is beneficial but remains challenging. Previously, generative deep learning (DL) model was implemented for de novo designing high diversified promoters[5, 6], suggesting combined DL models may

provide solutions for controllable design active promoters. However, it is still necessary to further explore the content of using DL models for de novo promoter design, which aims to address previous limitations and increase the available strong promoters.

Please see Line 45-55 in Page 3 in the revised manuscript.

(2) We are sorry for the confusion. The content for factors affected promoter activity has been rewritten as your suggestions, as follows:

Understanding the mechanism behind promoter activity has contributed to promoter research and design. Fundamental researches have suggested that promoter activity is mainly affected by functional motifs accommodation[7], RNA polymerase occupancy[8], and sequential fragments distribution.[9]

Please see Line 56-69 in Page 3 in the revised manuscript.

2.

Although the -35 and -10 motifs are important elements in many E. coli promoters, there are other important motifs in non-sigma70 promoters. For example, the motifs for sigma54 could be found in DOI: 10.1093/nar/gky010.

If non-sigma70 promoters were included in the training, validation, and testing data, then the number of promoters for each sigma factor should be clearly indicated, and the performance metrics for each sigma factor class should be provided separately.

Response: Thanks for your suggestion.

The promoters deposited in RegulonDB and Thomason et al organized database (we named NDB in the article) included all types of promoters, the classification and predictions based on different promoter types were added for RegulonDB, while for Thomason et al organized database, we adopted SAPPHIRE for the classification of sigma70 and non-sigma70 promoters. The corresponded prediction results were added to the Results section, as follows:

The model performance on different type of promoters was analyzed. RegulonDB[32] and NDB[34] both contain all types of promoters. Sigma70 promoter is the vast majority of sigma promoters existed naturally. Due to the promoters in NDB were not classified, we adopted SAPPHIRE[36] to predict sigma70 and non-sigma70 promoters in NDB (Supplementary File 1). Sigma70 promoter took the major portion for the two datasets, which is 48.5% for RegulonDB and 37.4% for NDB (Figure S5, Supporting Information). The best leads from the 10-fold cross-validation of PromoS and PromoA were used. We showed PromoS varied its accuracy from 0.74 to 0.81 for different types of promoters, whereas PromoA displayed similar PCC for sigma70 (PCC = 0.314) and non-sigma70 (PCC = 0.306) promoters (Figure S5, Supporting Information).

Please see Line 140-150 in Page 7 in the revised manuscript.

Figure S5 Classification and prediction accuracy against two datasets
 The promoter classification of RegulonDB[4] (A) and NDB[2] (B). PromoA and PromoS accuracy against NDB (C) and RegulonDB (D).

3.

The literature review is insufficient. In terms of promoter classification, there are quite a few DL methods which could achieve better performance than the ones cited by the authors.

In addition, there are other DL methods that focus on generating novel promoters. Here is an example: doi: 10.1016/j.cels.2020.05.007. Please provide a description of how this study differs from previous similar studies, as well as the unique methods employed, which contribute to the novelty of the results and findings. Could it be possible to conduct a benchmark to demonstrate the merits of this study compared with other available approaches?

Response: Thanks for your suggestion.

We have added changes including literature review and novelty to Introduction and Discussion sections, as follows:

Introduction:

Bioinformatics provided potential solutions for accurate and controllable designing of promoters compared with classic methods relied on mutagenesis or engineering the functional motifs.[7, 20-22] Growing number of DL methods were brought out for promoter strength prediction[19, 23-25] and generating of novel promoters[5, 26]. However, many of the architected models were not adopted for practical engineering

active promoters. Wang et al.[5] adopted a supervised model for predicting promoter transcription levels and generating promoters using a generative adversarial network (GAN). The generated promoters with predicted high activity were experimentally validated and aligned with a reported natural existed strong constitutive promoter J23119[27]. Most of the validated promoters displayed very low activity, suggesting the needs for further exploration for controllable designing active promoters.[5] GAN was also implemented for the architecting of DENs (developed deep exploration networks) brought out by Seelig et al.[6], which optimized its capability for engineering desired promoter sequences. These studies demonstrated that the GAN-based generative network can be used for designing of novel promoters, however, the combinatory uses to maximize the capability of DL methods for promoter design still needs further research.

This study aims to architect tools for de novo design active promoters with predictive strength. We brought out two approach, named “DRSAdesign” and “Ndesign” (Figure 1). DRSAdesign for de novo design promoters relied on generative network and strength prediction tools.[5] We attempted diffusion model-based generative network to integrate DRSAdesign, which aims to address several limitations of previous generative models, such as GAN is unstable during training and Variational AutoEncoder is hard to generate high quality samples.[28, 29] Taking advantage of promoter prediction models[19, 23, 24], we developed tools to adapt promoter length at 50 bp as input to predict their real or fake, and strength. Notably, DRSAdesign is a top-down process, in which generated promoters should be predicted as real promoter and then predicted their strength. This process aimed to cope with single model may not provide sophisticated results. Simultaneously, constraint-based generating sigma70 promoters was conducted to train our architected supervised model, and the model was further used to predict the novel generated promoter strength, this method we named Ndesign (Figure 1).

Please see Line 76-106 in Page 4 in the revised manuscript.

Discussion:

This study was carried out by attempting DL models for E. coli promoter design. We brought out theory and practice to validate the capability the DL tools. Previous architected DL tools showed high accuracy in promoter recognition and strength prediction.[19, 44-46] However, it is still lack of experimental evaluation of these tools. In this study, we built DRSAdesign by integrating generative network and supervised models inspired by previous studies.[5, 6] To address the limitations including the positive rates of the generated promoters, accuracy of supervised models, and stabilization of generative network, we attempted promoter real/fake prediction, optimization of activity prediction model, and using diffusion model (suggested more stabilized than GAN[28, 29]) for generating novel promoters in this study. Even 90% of the generated promoters were active, the generated promoters varied their activity by more than 150-fold, regardless they were all predicted as strong promoters (Figure 3D and Figure 4A), suggesting the weak performance of promoter strength prediction tools despite of their high in silico accuracy (Figure 2 and Figure 3).

Please see Line 285-298 in Page 12 in the revised manuscript.

4.

DRSAdesign is mentioned in Abstract, Results, and Discussion, but the term could not

be found in the figures of the main text. It is also confusing how DRSAdesign and Ndesign are different, and how the whole pipeline is put together, from model training to novel promoter generation. An overview diagram might help the audience to quickly grasp the whole idea of this study.

Response: Thanks for your suggestion.

We have added relevant description of the two design methods in Introduction, and Results, and Discussion sections, as follows:

Introduction:

This study aims to architect tools for de novo design active promoters with predictive strength. We brought out two approach, named “DRSAdesign” and “Ndesign” (Figure 1). DRSAdesign for de novo design promoters relied on generative network and strength prediction tools.[5] We attempted diffusion model-based generative network to integrate DRSAdesign, which aims to address several limitations of previous generative models, such as GAN is unstable during training and Variational AutoEncoder is hard to generate high quality samples.[28, 29] Taking advantage of promoter prediction models[19, 23, 24], we developed tools to adapt promoter length at 50 bp as input to predict their real or fake, and strength.

Please see Line 93-100 in Page 5 in the revised manuscript.

Results:

By extracting predicted real promoters (using PromoR) from the generated promoters showed unexpected logos were significantly minimized (represented as “Real” in Figure 3C). The predicted real promoters were predicted their strength for experimental validation. The combined diffusion model and supervised models for generating promoters and predicted their activity named DRSAdesign.

Please see Line 199-203 in Page 9 in the revised manuscript.

Discussion:

In this study, we built DRSAdesign by integrating generative network and supervised models inspired by previous studies.[5, 6]

Please see Line 289-290 in Page 12 in the revised manuscript.

Figure 1

5.

No clear descriptions about the rationale and functionality of the "unconditional diffusion model", or "PromoDiff", could be found. There could be other choices of DL models for creating a generative method, such as GAN, autoencoder, etc. Please explain.

Besides, in-depth statements about the design, and the working process of this diffusion model must be given, and such as the flow of x_0 to z .

Response: Thanks for your suggestion.

To clarify the unconditional diffusion model and why we selecting diffusion model, we added description in the Introduction, Results, and Experimental section, as follows:

Introduction:

We attempted diffusion model-based generative network to integrate DRSAdesign, which aims to address several limitations of previous generative models, such as GAN is unstable during training and Variational AutoEncoder is hard to generate high quality samples.[28, 29]

Please see Line 96-99 in Page 5 in the revised manuscript.

Results:

We introduced PromoDiff, an unconditional diffusion model-based network for end-to-end generating promoters by learning features from natural promoters. In this case, unconditional promoter generation indicates that the model converts noise into any random representative data sample without any guide. Thus, the unconditional model can generate a promoter of any nature. The use of diffusion model on promoters is to gradually add Gaussian noise to the input data-point (X_0) by a series of steps (Z), and the model learnt to reverse the noising process which can recover the sample to the denoised state (Figure 3A). Diffusion model is basically integrated by UNet for extracting features from real samples and generating samples mimic the input ones with variations [28].

Please see Line 172-181 in Page 8 in the revised manuscript.

Experimental section:

PromoDiff is a diffusion model integrated by UNet. UNet is created for semantic segmentation, which contains an encoder and decoder block for feature extraction and sample size recovery purpose. To learn features of promoters, we used ResNet and selfattention on the encoder side, while to recover the promoter size we adopted upsampling on the decoder side. For the process of diffusion model[28], an input promoter was extracted its features using one-hot encoding method and the data-point represented as X_0 . The forward diffusion process is gradually adding Gaussian noise to X_0 through T steps, while the Gaussian noise with variance can produce a new latent variable X_t . Then, the architected network is trained to recover the original data by reversing the noising process, indeed, the reverse diffusion process can generate new data which highly mimic the original ones.

Please see Line 434-444 in Page 18 in the revised manuscript.

It is possible that the diffusion model outputs real/known promoters (i.e. identical to training promoter sequences). It is necessary to explicitly address how such replicates are removed, the proportion of novel promoters in all generated promoters, and the extent of how the novel promoters are different in sequence contents from the real/known promoters.

Response: Thanks for your suggestion.

We have consulted if replicates existed in the generated promoters, and shown nonredundant promoters were generated, the original file for the generated promoters were uploaded as “Supplementary File 2” in the revised version.

The sequential difference of the generated promoters and the training set was addressed, the changes as follows:

We optimized the learning rate according to RPP, which showed promoters generated within epoch 600 to 800 with the highest RPP (Figure S8, Supporting Information). In addition, epoch 620 showed the highest RPP and the SL was less chaotic (Figure S9, Supporting Information). We further generated 25000 promoters adopting epoch 620, visualization of the SL revealed these promoters have the same -10 motif (TATAAT) as the training samples, but the -35 motif varied from TT to AT or TA (represented as “All” in Figure 4). Meanwhile, generated promoters with unexpected logos aligned with the training set, which showed high “GC” content within -50 to -35 region and high “A” content within -10 to 0 region. Notably, the generated promoters showed non-redundant with the training set (Supplementary File 2).

By extracting predicted real promoters (using PromoR) from the generated promoters

showed unexpected logos were significantly minimized (represented as “Real” in Figure 3C).

Please see Line 189-201 in Page 8 in the revised manuscript.

6.

Please explain what J23119 is.

Response: We are sorry for the confusion.

The explanation added in the Introduction section, as follows:

The generated promoters with predicted high activity were experimentally validated and aligned with a reported natural existed strong constitutive promoter J23119[27].

Please see Line 83-85 in Page 4 in the revised manuscript.

7.

In "Experimental section", please justify the usage of "egfp" as a sufficient validation platform for assessing promoter strength. What is egfp? Is this term already used by the cited reference (doi: 10.1002/iub.1041)?

Response: Thanks for your suggestion.

The term of egfp is used in the reference, as below:

IJMB.Life, Author manuscript; available in PMC 2013 Aug 1.
Published in final edited form as:
IJMB.Life. 2012 Aug; 64(8): 684-689.
Published online 2012 May 28. doi: 10.1002/iub.1041

PMCID: PMC3415465
NIHMSID: NIHMS394178
PMID: 22639380

Fluorescent protein engineering by *in vivo* site-directed mutagenesis

Melvys Valledor-Cabañes,[✉] Qinghua Hu,[§] Paul Schiller,[§] and Richard S. Myers[✉]

• Author information • Copyright and License information • Disclaimer

The publisher's final edited version of this article is available free at IJMB.Life

Associated Data

• Supplementary Materials

Summary

Go to: ▶

In vivo site-directed mutagenesis by ssDNA recombineering is a facile method to change the color of fluorescent proteins without cloning. Two different starting alleles of GFP were targeted for mutagenesis: *gfpmut3** residing in the *E. coli* genome and *egfp* carried by a bacterial/mammalian dual expression lentiviral plasmid vector.

Plasmids

pDual-**eGFP** was created by ligating a PCR product (produced using 5' phosphorylated primers (oligos 61 and 62) to amplify the T7 promoter region from pET28a) to SmaI digested pNL-**eGFP**/CEF [11] and screening transformed Rosetta-gami²(DE3) colonies for gain of green fluorescence. The inserted sequence contains the T7 promoter, the LacI binding site (Lac operator), a Shine-Dalgarno sequence, a His₆ tag and a T7 tag. The resulting His₆-T7 tag **eGFP** fusion protein is strongly expressed from pDual-**egfp** in the Rosetta-gami²(DE3) and the construct was verified by sequencing using oligo 44 as a primer. pDual-**eGFP**(W₆₆) and pDual-**eGFP**(Stop₆₆) were created from pDual-**eGFP** by recombineering with oligos 59 and 60 respectively in strain RIK473. pDual-**eGFP**(H₆₆) and pDual-**eGFP**(Y₆₆) were created from pDual-**eGFP** using the QuikChange Lightning Site-Directed Mutagenesis Kit (Agilent Technologies) with oligos 68-71. Mutagenesis was confirmed by allele-specific PCR (Y₆₆ with oligos 49 and 45 or oligos 72 and 46, W₆₆ with oligos 50 and 45, Stop₆₆ with oligos 53 and 45, H₆₆ with oligos 73 and 46, T₂₀₃ with oligos 45 and 74, Y₂₀₃ with oligos 45 and 75) and sequencing using oligos 45 and 46. pCMV-VSV-G and pCMV-dRR.2dvpr used to create lentivirus transducing particles were gifts from Dr. Parya Rai. All oligo sequences are listed in Supporting Table 2 online.

The significance of using egfp as validation platform was added in the Results section, as follows:

Green fluorescent protein (GFP) was widely adopted for characterizing promoter strength previously [5, 39]. In this study, we used an engineered EGFP (enhanced GFP) with stronger fluorescent signal to evaluate the strength of generated promoters [40]. We selected promoters with PromoS scores above 0.9 and PromoA scores top-ranked 50 for inducing egfp expression.

Please see Line 204-208 in Page 9 in the revised manuscript.

8.

In "Dataset". Please explain what are fake promoters. Why they are fake? What is the purpose of using fake promoters in the training?

Response: We are sorry for the confusion.

The explanation for fake promoter and their uses for training was added to Results section, as follows:

PromoR was developed for predicting real and fake promoters. The purpose for training

PromoR is to understand the key elements consist of real promoters and avoid of designing fake but high active promoters. The fake promoters represented nonpromoter sequences without promoter activities as annotated in RegulonDB[32], which were obtained from the middle regions of long coding sequences and convergent intergenic regions in *E. coli* K-12 genome.[37]

Please see Line 153-158 in Page 7 in the revised manuscript.

9.

In "Dataset". It appears that the authors acquired > 10,000 promoters from NDB, on which I cannot find the resource name from the cited references. Besides, the authors must explain explicitly how to confirm that all promoters contain -35 and -10 motifs. Based on the reference of the plausible source of NDB (doi:10.1128/jb.02096-14), the statements about the promoters include "The majority of pTSS, iTSS, and asTSS are preceded by σ 70 promoter elements", and "The enrichment for two T residues comprising a potential σ 70 -35 element was significantly less than what was observed for the -10 element".

Response: We are sorry for the confusion.

(1) NDB is a short name that we created in this manuscript for convenient uses, to avoid the misunderstanding, we made changes in the Results section, if that is not proper we could fix it further, as follows:

The dataset contains *E. coli* native promoters obtained by global TSS maps created by Thomason et al.[34], which we named NDB for convenient cited in this study.

Please see Line 129-130 in Page 6 in the revised manuscript.

(2) Our original description for the dataset created by Thomason et al. was not correct, these promoters were identified by mapping 14868 potential TSS in *E. coli* genome (doi:10.1128/jb.02096-14), and the sequences we actually obtained from the study of Wang et al. conducted (doi: 10.1093/nar/gkaa325), their original description about the dataset as below:

We predicted a total of 14,868 potential TSS mapping throughout the *E. coli* genome (see Data set S1 in the supplemental material). Of these, 6,297 were detected under all three conditions, 1,151 were detected only in cells growing exponentially in M63 minimal medium, 470 TSS were found in cells growing exponentially in LB, and 1,947 were found in stationary-phase cells growing in LB (Fig. 2A; see also Fig. S3A in the supplemental material for examples of TSS detected under only one condition). The higher number of TSS identified for the LB stationary-phase cells might be a result of changes in transcriptional programs required to survive in the stationary phase (38).

Thomason et al.

The model training of predictive models

For the first round of preselection, we trained a convolutional neural network (CNN) predictive model based on public transcriptome data. The training dataset was from Thomason *et al.*, which contains 14098 promoters with corresponding gene expression levels measured by dRNA-seq

Wang et al.

Thomason et al. illustrated these sequences have promoter function which can induce transcription, but may not contain a regularized -35 and -10 motif. The -35 and -10

distribution was predicted using SAPPHERE (doi:10.1186/s12859-020-03730-z). We made changes in the Experimental section as follows: NDB contains 11,884 non-redundant samples which were 50 bp upstream of the TSS [5, 34]. Noted that 6,297 of the promoters were constitutive promoters, while the others functioning on certain cell phases. The -35 and -10 elements prediction was conducted using SAPPHERE [36], which showed only 4442 promoters with standard sigma70 -35 and -10 motifs (Supplementary File 1).

Please see Line 387-391 in Page 16 in the revised manuscript.

10.

In "Network architecture". PromoA is trained to perform a transcription-level prediction, and PromoR is trained to classify promoters. Both models were built by including an architecture consisting of ResNet and Attention layers. However, it is suspicious if PromoR can really achieve the accuracy level required for selecting novel synthetic promoters, and a thorough benchmark to compare the it with other tools such as CNNProm, IPromoter-2L, etc. might further elucidate the value of these two new models.

Response: We are sorry for the confusion.

The three supervised networks including PromoR, PromoA, and PromoS were built to predict a sequence at length of 50 bp, which is used to fit the generated promoter length by the diffusion model. We showed the accuracy of PromoR was similar as iPromoterCLA using the same 5-fold cross validation on the same dataset. PromoR was then trained using a larger dataset by adding promoters from the dataset offered by Thomason et al. and randomly retrieved from the non-promoter sequences. The retrained PromoR showed higher accuracy than iPromoter-CLA in every validated aspects. However, further benchmark test was not able to carry out since (1): the test dataset adopted by previous study with a lot of redundant data (test set organized by Liu et al., doi: 10.1093/bioinformatics/btx579, previous independent test carried out as DOI: 10.1093/bioinformatics/btx579; doi: 10.1016/j.cmpb.2022.107087; doi: 10.1016/j.ygeno.2018.12.001), and we provided a separate file to clarify this (Supplemented file 1 for reviewers), (2): when we try to retrieve a 50 bp length promoter from the independent dataset, we found the retrieved region is not consistent within a steady region, such as 19-69, this makes the retrieved region different from what we retrieved from RegulonDB.

In addition, it is confusing why one attention layer is placed after each set of ResBlocks in PromoA/S/R. Attention layer is different from self-attention layer in that the former is to learn the dependence between each input token and the output classification. One attention might be sufficient for the classification task.

Response: We are sorry for the confusion.

This work we used self-attention layer, not attention layer, we have fixed this error throughout the manuscript, including the main text and the correlated figures as below. The self-attention block integrated this network was optimized its insertion position according to the accuracy, and we attempted to insert only one self-attention layer after each CNN block as well as inserting two or three self-attention layer after the CNN block, but these integrating methods gave lower accuracy compared with using four self-attention layer after each CNN block. To clarify this, an extended explanation was added to the Results section as follows:

We attempted sequential feature extraction using one-hot encoding strategy, pseudodinucleotide composition (pseDNC)[24], and the combined methods (Figure S1, Supporting Information). Meanwhile, we attempted Convolutional Neural Network (CNN), BiLSTM, ResNet, and self-attention for the network construction. The network was trained using the promoter at a length of 50 bp, and optimized the architecture based on the 10-fold cross-validation accuracy. We showed one-hot method for feature extraction, and integrating the network using ResNet and self-attention provided better performance which achieved 0.7806 (Figure 2A and Figure S2, Supporting Information). Training different model architectures using the promoters at a length of 81 bp also supporting that integrating ResNet and self-attention can contribute to better performance using 10-fold cross-validation (Figure 2B and Figure S3, Supporting Information). The model robustness was validated by comparing with previous state-of-art method trained with longer promoters.[19] We showed PromoS trained with shorter promoters can still achieve accurate predictions (Figure 2C).

Please see Line 112-126 in Page 5 in the revised manuscript.

Figure 2

Figure S2 Optimization of deep learning model

Deep learning model PromoS architected using different blocks and their accuracy for predicting strong/weak promoters were shown. Noted CNN represented Convolutional Neural Network, BiLSTM represented Bidirectional-LSTM, and SA represented selfattention. The accuracy achieved using 10-fold cross validation.

PromoR and PromoS share the same architecture at the front-end. Subsequently, they individually classify input sequences as either Strong or Weak, and either Real or Fake. Since the front-end models are shared, for each input they give respective outputs. Thus, there is a question about how to resolve this case: Fake promoter is Strong or Weak.

Response: We are sorry for the confusion.

In previous models, such as iPSW (doi: 10.1016/j.ygeno.2019.08.009) and iPromoterCLA (doi: 10.1016/j.cmpb.2022.107087), they used a same network for firstly

predicting the real/false of given promoter, and then predicted their strong/weak. Like below:

- I am not clear on why certain datasets were used for certain models. Some models use just RegulonDB, some use NBD as well, etc. Why would the maximum amount of available data not be used in every model?

Response: We are sorry for the confusion.

RegulonDB only contained promoter strong/weak, and real/fake information. NBD only contain promoter transcription level information. Meanwhile, sequences in RegulonDB and NBD were at a length of 81 bp and 50 bp respectively. These two dataset are used for different training purpose. To avoid confusion, an explanation has been added to the Results section, as follows:

This network was trained using RegulonDB[32] by adjusting the promoter length to 50 bp. We attempted sequential feature extraction using one-hot encoding strategy, pseudodinucleotide composition (pseDNC)[24], and the combined methods (Figure S1, Supporting Information). Meanwhile, we attempted Convolutional Neural Network

(CNN), BiLSTM, ResNet, and self-attention for the network construction. The network was trained using the promoter at a length of 50 bp, and optimized the architecture based on the 10-fold cross-validation accuracy. We showed one-hot method for feature extraction, and integrating the network using ResNet and self-attention provided better performance which achieved 0.7806 (Figure 2A and Figure S2, Supporting Information). Training different model architectures using the promoters at a length of 81 bp also supporting that integrating ResNet and self-attention can contribute to better performance using 10-fold cross-validation (Figure 2B and Figure S3, Supporting Information). The model robustness was validated by comparing with previous state-of-art method trained with longer promoters.[19] We showed PromoS trained with shorter promoters can still achieve accurate predictions (Figure 2C).

Please see Line 111-126 in Page 5 in the revised manuscript.

The dataset contains E. coli native promoters obtained by global TSS maps created by Thomason et al.[34], which we named NDB for convenient cited in this study. The database contains 11,884 non-redundant promoters, while 3098 of them are not constitutive promoters.[34] Regarding the sample diversity and complexity, PromoA achieved a Pearson correlation coefficient (PCC) of 0.31 by 10-fold validation (Figure 2D and Figure S4, Supporting Information), which was slightly higher than previously reported 0.25.[5]

Please see Line 129-135 in Page 6 in the revised manuscript.

PromoR was developed for predicting real and fake promoters. The purpose for training PromoR is to understand the key elements consist of real promoters and avoid of designing fake but high active promoters. The fake promoters represented nonpromoter sequences without promoter activities as annotated in RegulonDB[32], which were obtained from the middle regions of long coding sequences and convergent intergenic regions in E. coli K-12 genome.[37]

Please see Line 153-158 in Page 7 in the revised manuscript.

- The authors state that one innovation in this work is "characterizing key promoter elements" (from the abstract). Where was this discussed? I was expecting a more extensive analysis of sequence determinants of promoter activity but most discussion was just related to the well-established -10 box, -35 box, and spacer, and any new knowledge discovered in this work is unclear to me.

Response: We are sorry for the confusion.

We have revised the Abstract, Results, and discussion sections for better presentation. To make it clear of how these promoter elements can actually affect promoter function, we mainly added description in the Discussion section. All as follows:

Abstract:

Taking advantages of the organized DL models in this work, possible 6-mers as key functional motifs of sigma70 were predicted, suggesting promoter recognition and strength mainly relying on the functional motifs' accommodation.

Please see Line 34-36 in Page 2 in the revised manuscript.

Results:

Identification of random 6-mer as potential functional motif for sigma70 promoter
Up to 99.1% of constraint-based generated random promoters were active, highlighting the significance of functional motifs. In addition to design sigma70 promoter, we adopted PromoR and PromoS for further predicting potential 6-mers that may be useful

for constructing sigma70 promoters. Firstly, we maintained the conserved -35 motif (TTGACA) to investigate possible 6-mer that can consist the -10 motif.

Please see Line 259-264 in Page 11 in the revised manuscript.

Discussion:

Based on PromoR prediction, 23.5% of the random sequences carried all possible 6-mers as the -10 motif and the conserved -35 motif (TTGACA) has a 90% chance to form real promoters (Supplementary File 4). In addition, 100% of the random sequences carried all possible 6-mers as the -35 position and the conserved -10 motif (TATAAT) can form real promoters (Supplementary File 4). These results indicated PromoR recognizes certain 6-mers as a critical standard to evaluate if the given sequences can be real promoters.

Please see Line 321-327 in Page 14 in the revised manuscript.

- Line 72 - the authors cite previous work generating novel promoters with deep learning - but I thought that was the innovation of this work over previous work according to their abstract. Can the authors please clarify their innovation?

Response: Thanks for your suggestion.

We have rewritten the Introduction section to clarify the novelty of this work, as follows: This study aims to architect tools for de novo design active promoters with predictive strength. We brought out two approach, named “DRSAdesign” and “Ndesign” (Figure 1). DRSAdesign for de novo design promoters relied on generative network and strength prediction tools.[5] We attempted diffusion model-based generative network to integrate DRSAdesign, which aims to address several limitations of previous generative models, such as GAN is unstable during training and Variational AutoEncoder is hard to generate high quality samples.[28, 29] Taking advantage of promoter prediction models[19, 23, 24], we developed tools to adapt promoter length at 50 bp as input to predict their real or fake, and strength. Notably, DRSAdesign is a top-down process, in which generated promoters should be predicted as real promoter and then predicted their strength. This process aimed to cope with single model may not provide sophisticated results. Simultaneously, constraint-based generating sigma70 promoters was conducted to train our architected supervised model, and the model was further used to predict the novel generated promoter strength, this method we named Ndesign (Figure 1).

Please see Line 93-106 in Page 5 in the revised manuscript.

- The phrase "unconditional generating promoters" (line 77) is completely unclear to me, and resurfaces later as well. What does "unconditional" mean in this context?

Response: We are sorry for the confusion.

An explanation for unconditional generating promoters were added to the Results and Experimental sections, as follows:

We introduced PromoDiff, an unconditional diffusion model-based network for end-to-end generating promoters by learning features from natural promoters. In this case, unconditional promoter generation indicates that the model converts noise into any random representative data sample without any guide. Thus, the unconditional model can generate a promoter of any nature.

Please see Line 172-176 in Page 8 in the revised manuscript.

- Line 157: What is UNet? This is the only place in the paper where I see it mentioned.

Response: We are sorry for the confusion.

An explanation for UNet were added to the Experimental sections, as follows:
UNet is created for semantic segmentation, which contains an encoder and decoder block for feature extraction and sample size recovery purpose. To learn features of promoters, we used ResNet and self-attention on the encoder side, while to recover the promoter size we adopted upsampling on the decoder side.

Please see Line 434-438 in Page 18 in the revised manuscript.

- Line 76: The authors begin using promoter J23119 without discussing what it is or why it's useful as a reference.

Response: We are sorry for the confusion.

The explanation for J23119 and its significance were added in the Introduction section, as follows:

The generated promoters with predicted high activity were experimentally validated and aligned with a reported natural existed strong constitutive promoter J23119[27].

Please see Line 83-85 in Page 4 in the revised manuscript.

- I do not understand the point on line 265 at all. What is "high regularity data"? Is the high performance mentioned not just the result of training a model to memorize a specific dataset? If $R = 0.94$ was possible already, it would completely invalidate the need for the submitted work.

Response: We are sorry for the confusion.

The work conducted by Zhao et al. (doi.org/10.1021/acssynbio.1c00117) actually mutates an input promoter sequence, and most of the promoter they tested with a single or double mutation based on the input sequence, which made the training sequences very high similarity, and easy to be learnt their activity. To avoid confusions, the Discussion section, as follows:

Even though PromoNet trained with constraint-based promoters gave higher PCC, however, less training data may limited the performance of PromoNet for the exploration of sequential possibilities except the -35 and -10 motifs.

Please see Line 341-343 in Page 15 in the revised manuscript.

Reviewer #3:

"In this study, Zhou et al. developed deep learning pipelines for generating, recognizing, and characterizing promoter strengths. They also experimentally validated the performance of their model, and identified several stronger promoters compared to the positive control, making it an intriguing and useful study. However, I have several comments that could potentially enhance the manuscript's quality.

Response: Thanks for your positive comment. We have carefully read all the comments, and the manuscript has been revised as suggested.

Major:

1.

Abstract: In the abstract, the authors claim that deep learning tools for promoter design and characterization still lack research. However, they cite several previously published models for promoter design and characterization. It would be helpful to discuss the differences and explain the uniqueness of this paper. Additionally, in the Discussion section, the authors mention that GAN or diffusion models mostly generate low-activity promoters based on their training dataset, while their deep learning model, PromoS, with over 20,000 training data points, cannot achieve higher activity promoters. They further validated that a simple CNN model with constraint-based random promoter

generation, PromoNet, is able to accomplish this. Therefore, it is important to justify the claim of lacking deep learning models and clarify the specific benefits and advancements presented in this paper.

Response: Thanks for your suggestion.

We have revised the Abstract and Results sections to explain the uniqueness of our paper, as follows:

Abstract:

Here, we explored the content of controllable designing active E.coli promoter by combining multiple deep learning models. Firstly, we brought out “DRSAdesign”, which relied on diffusion model to generate novel promoters, followed by predicting their real/fake and strength. Experimental validation showed 45 out of 50 generated promoters were active with high diversity, but most promoters with relative low activity. Next, “Ndesign” was brought out relied on generating random sequences carried functional -35 and -10 motifs of sigma70 promoters, and predicted their strength using our architected DL model. The DL model was trained and validated using our generated 200 and 50 promoters, which displayed a Pearson Correlation of 0.49 and 0.43, respectively.

Please see Line 25-34 in Page 2 in the revised manuscript.

We have revised the Discussion section to bring out “justify the claim of lacking deep learning models and clarify the specific benefits and advancements presented in this paper”, as follows:

This study was carried out by attempting DL models for E. coli promoter design. We brought out theory and practice to validate the capability the DL tools.

Please see Line 285-286 in Page 12 in the revised manuscript.

On the other side, splitting the promoters in RegulonDB and NDB showed the constitutive promoters only took a portion of less than half in the given dataset, while the other promoters contain other sigma types or unknown active promoters (Figure S4, Supporting Information), indicating lack of sophisticated DL models to capture features from the variety of dissimilar data. Moreover, the portion of sigma70 promoter in NDB is significantly less than that in RegulonDB, and regression prediction is harder than binary prediction which together contributed to the low accuracy of PromoA.

Please see Line 301-308 in Page 13 in the revised manuscript.

2.

Line 96: Please mention the reason for choosing a length of 50 bp for PromoS training.

Response: Thanks for your suggestion.

Training the model of PromoS with a sequence length of 50 bp is for fitting the generated promoters, the generated promoter is from the diffusion model organized in this work, and the protocol brought out in our study as “DRSAdesign”. To avoid confusions, we have made changes in Introduction and Results sections, as follows:

Introduction:

This study aims to architect tools for de novo design active promoters with predictive strength. We brought out two approach, named “DRSAdesign” and “Ndesign” (Figure 1). DRSAdesign for de novo design promoters relied on generative network and strength prediction tools.[5] We attempted diffusion model-based generative network to integrate DRSAdesign, which aims to address several limitations of previous generative models, such as GAN is unstable during training and Variational

AutoEncoder is hard to generate high quality samples.[28, 29] Taking advantage of promoter prediction models[19, 23, 24], we developed tools to adapt promoter length at 50 bp as input to predict their real or fake, and strength.

Please see Line 93-101 in Page 5 in the revised manuscript.

Results:

By extracting predicted real promoters (using PromoR) from the generated promoters showed unexpected logos were significantly minimized (represented as “Real” in Figure 3C). The predicted real promoters were predicted their strength for experimental validation. The combined diffusion model and supervised models for generating promoters and predicted their activity named DRSAdesign.

Please see Line 199-203 in Page 9 in the revised manuscript.

3.

Please explain and discuss why multiple models (PromoR/S/A) were developed instead of a single model that can accomplish all these tasks.

Response: We are sorry for the confusion.

Using of multiple models is adopted as a way to avoid single models may provide not trustable results. We added description in our Introduction section, as follows:

Notably, DRSAdesign is a top-down process, in which generated promoters should be predicted as real promoter and then predicted their strength. This process aimed to cope with single model may not provide sophisticated results.

Please see Line 101-104 in Page 5 in the revised manuscript.

4.

Line 125: Please provide additional explanation about why the weak performance on NDB leads to the conclusion of weak performance on diversified samples. Furthermore, please clarify why the NDB database is considered more diversified compared to RegulonDB.

Response: Thanks for your suggestion.

The explanation has been added in the Results and Discussion section, as follows:

Results:

The model performance on different type of promoters was analyzed. RegulonDB[32] and NDB[34] both contain all types of promoters. Sigma70 promoter is the vast majority of sigma promoters existed naturally. Due to the promoters in NDB were not classified, we adopted SAPPHIRE[36] to predict sigma70 and non-sigma70 promoters in NDB (Supplementary File 1). Sigma70 promoter took the major portion for the two datasets, which is 48.5% for RegulonDB and 37.4% for NDB (Figure S5, Supporting Information).

Please see Line 140-146 in Page 7 in the revised manuscript.

Discussion:

On the other side, splitting the promoters in RegulonDB and NDB showed the constitutive promoters only took a portion of less than half in the given dataset, while the other promoters contain other sigma types or unknown active promoters (Figure S4, Supporting Information), indicating lack of sophisticated DL models to capture features from the variety of dissimilar data. Moreover, the portion of sigma70 promoter in NDB is significantly less than that in RegulonDB, and regression prediction is harder than binary prediction which together contributed to the low accuracy of PromoA.

Please see Line 301-308 in Page 13 in the revised manuscript.

Figure S5 Classification and prediction accuracy against two datasets
 The promoter classification of RegulonDB[3] (A) and NDB[1] (B). PromoA and PromoS accuracy against NDB (C) and RegulonDB (D).

5.

Line 126: What is the source of the 11,884 non-promoters mentioned?

Response: We are sorry for the confusion.

We added the description in the Results section, as follows:

The 11,884 non-promoters were obtained by randomly retrieving 50 bp from the 81 bp of non-promoter sequences in RegulonDB[32].

Please see Line 165-167 in Page 7 in the revised manuscript.

6.

Since the training dataset is derived from E. coli, is this a deep learning tool specifically designed for E. coli promoters? If so, please mention this in the abstract and the main text to avoid confusion.

Response: Thanks for your suggestion.

The clarification has been added in the Abstract and Introduction section, as follows:

Title:

Deep learning-assisted designing novel promoters in Escherichia coli

Abstract:

Here, we explored the content of controllable designing active E.coli promoter by combining multiple deep learning models.

Please see Line 25-26 in Page 2 in the revised manuscript.

Discussion:

This study was carried out by attempting DL models for E. coli promoter design. We brought out theory and practice to validate the capability the DL tools.

Please see Line 285-286 in Page 12 in the revised manuscript.

7.

What are the criteria used to define a strong promoter compared to a weak promoter in

the training dataset of PromoS?

Response: We are sorry for the confusion.

Promoter strong/weak is actually defined by RegulonDB using the confidence for all evidence sources. The strong and weak term was brought out firstly in RegulonDB (version 6.0) (doi: 10.1093/nar/gkm994), the cited paragraph as below:

The evidences associated to all RegulonDB objects are now classified as 'strong' or 'weak,' based on the confidence level of the experiment or prediction that supports objects and their relationships. A 'strong' evidence is assigned to an object when the experimental data provide high certainty of its existence; otherwise, it is a 'weak' evidence. Examples of strong evidences are DNA binding of purified TF for regulatory interactions, mapping of TSSs for promoters, and length of mRNA for transcription units. On the other hand, gene expression analyses and computational predictions are considered weak evidences. It is important to state that several weak evidences for an object do not become a strong one. These two types of evidences are distinguished graphically with solid or dashed lines for objects supported by strong or weak evidences, respectively.

Minor:

1. Line 43-44: Consider revising the sentence "It is important for engineering strong promoters for the overexpression of recombinant proteins or optimization of metabolic engineering in microorganisms" or explain the necessity of engineering if the promoter is already strong.

Response: Thanks for your suggestion.

The given sentence was removed, and the paragraph has been rewritten as below:

Controlling gene expression level is important for protein recombinant expression and engineering microbial cell factories.[2-4] However, there is limited number of naturally existing strong promoters to facilitate high level protein expression. Current techniques mainly relied on mutagenesis or shuffling key elements for screening novel promoters, which is labor-intensive. Controllable designing stronger is beneficial but remains challenging.

Please see Line 45-50 in Page 3 in the revised manuscript.

2. Line 180: "The active promoters with -35 and -10 motifs of TT and ATTTTA (Figure 4C)" is not a complete sentence. Please revise it accordingly.

Response: We are sorry for the confusion.

The sentence was removed to avoid confusion.

3. The author alternately uses "strength" and "activity" to refer to the ability of the promoter. It would be beneficial to maintain consistency in the terminology used.

Response: We are sorry for the confusion.

We have fixed this error throughout the manuscript.

4. Figure 4D: Please label the significance in the figure and consider adding a reference line or using different colors to differentiate between promoters with increased activity compared to the control and those without."

Response: Thanks for your suggestion.

We added a reference line to both positive and negative control in Figure 4 and 5 as below:

Figure 3

Figure 4

Reviewer #4:

The authors systematically studied the features of *E. coli* promoters, and architected diffusion model based deep learning model for generating new promoters. Meanwhile, the authors built deep learning models for predicting the strength of promoters, which assisted the characterization of promoters, and selecting of high activity promoters. This may be the first time for attempting using diffusion model for generating promoters, which is an interesting work. Moreover, this study tested the model accuracy regarding the architected deep learning models were previously achieved high in silico accuracy. There are issues with the readability and clarity of the results, which should be addressed, we recommend to accept the paper after revising these minor issues.

Response: Thanks for your positive comment. We have carefully read all the comments, and the manuscript has been revised as suggested.

1:

Line 95, "This network was trained using RegulonDB[19] by adjusting the promoter length to 50 bp". This may be confused to the authors of why adjusting the promoter length to 50 bp.

Response: We are sorry for the confusion.

Training the model of PromoS with a sequence length of 50 bp is for fitting the generated promoters, the generated promoter is from the diffusion model organized in this work, and the protocol brought out in our study as “DRSAdesign”. To avoid confusions, we have made changes in Introduction and Results sections, as follows:

Introduction:

This study aims to architect tools for de novo design active promoters with predictive strength. We brought out two approach, named “DRSAdesign” and “Ndesign” (Figure 1). DRSAdesign for de novo design promoters relied on generative network and strength prediction tools.[5] We attempted diffusion model-based generative network to integrate DRSAdesign, which aims to address several limitations of previous generative models, such as GAN is unstable during training and Variational AutoEncoder is hard to generate high quality samples.[28, 29] Taking advantage of promoter prediction models[19, 23, 24], we developed tools to adapt promoter length at 50 bp as input to predict their real or fake, and strength. Notably, DRSAdesign is a top-down process, in which generated promoters should be predicted as real promoter and then predicted their strength. This process aimed to cope with single model may not provide sophisticated results. Simultaneously, constraint-based generating sigma70 promoters was conducted to train our architected supervised model, and the model was further used to predict the novel generated promoter strength, this method we named Ndesign (Figure 1).

Please see Line 93-106 in Page 5 in the revised manuscript.

Results:

By extracting predicted real promoters (using PromoR) from the generated promoters showed unexpected logos were significantly minimized (represented as “Real” in Figure 3C). The predicted real promoters were predicted their strength for experimental validation. The combined diffusion model and supervised models for generating promoters and predicted their activity named DRSAdesign.

Please see Line 199-203 in Page 9 in the revised manuscript.

2:

Line 105, "Recent study illustrated gene expression was highly correlated with transcription in E. coli regardless of other factors". This is not accurate, the other factors showed minor influences to the trade-off of transcription and expression rather than "regardless".

Response: We are sorry for the confusion.

The given sentence was rewritten as below:

Recent study illustrated gene expression was highly correlated with transcription in E. coli [33].

Please see Line 127-128 in Page 6 in the revised manuscript.

3:

Line 112, "We further validated the downstream DNA's interruption to transcription". This sentence is confused.

Response: We are sorry for the confusion.

The given sentence was rewritten as below:

To test our hypothesis of whether the downstream area of promoter can interrupt the transcription, we further extended the promoter length to 81 bp, 100 bp, 150 bp, 200 bp,

300 bp, and 400 bp by adding certain length of downstream region according to E. coli K-12 genome.[35]

Please see Line 135-138 in Page 6 in the revised manuscript.

4:

Line 131, "The -35 motif is conserved which mostly starts with TT followed by less conserved GAGC in constitutive promoters". A reference is needed for this.

Response: We are sorry for the confusion.

This sentence is removed to avoid confusions.

5:

Line 118, "Identification of functional motifs for promoter architecture". This section mainly describe the possibilities of using random 6-mers as -35 and -10 motif to generate active promoters, while the spacer between -35 and -10 motif is 17 bp, therefore, this is a constraint-based generating promoters. The authors should reformatting this section to avoid confusions.

Response: We are sorry for the confusion.

The paragraph was rewritten as below:

Architected supervised model for promoter real and fake prediction

PromoR was developed for predicting real and fake promoters. The purpose for training PromoR is to understand the key elements consist of real promoters and avoid of designing fake but high active promoters. The fake promoters represented nonpromoter sequences without promoter activities as annotated in RegulonDB[32], which were obtained from the middle regions of long coding sequences and convergent intergenic regions in E. coli K-12 genome.[37] PromoR was initially trained using 3382 promoters and 3382 non-promoter sequences from RegulonDB[32] by adjusting the promoter length to 50 bp. Through 5-fold cross-validation, PromoR achieved an accuracy of 0.86 which is comparable to the reported highest values of 0.8603[19] (Supplementary Table S1). However, PromoR displayed an accuracy of 0.767 on NDB[34], suggesting its weak performance on diversified samples. To make PromoR a robust tool, a combined dataset consisting of RegulonDB, NDB, and 11,884 nonpromoters was organized (totally 30,532 samples). The 11,884 non-promoters were obtained by randomly retrieving 50 bp from the 81 bp of non-promoter sequences in RegulonDB[32]. PromoR trained with the combined dataset achieved an accuracy of 0.8861 using 5-fold cross-validation (Supplementary Table S1 and Figure S6, Supporting Information).

Please see Line 152-169 in Page 7 in the revised manuscript.

Identification of random 6-mer as potential functional motif for sigma70 promoter

Up to 99.1% of constraint-based generated random promoters were active, highlighting the significance of functional motifs. In addition to design sigma70 promoter, we adopted PromoR and PromoS for further predicting potential 6-mers that may be useful for constructing sigma70 promoters. Firstly, we maintained the conserved -35 motif (TTGACA) to investigate possible 6-mer that can consist the -10 motif. Random promoters were generated using an optimal spacer length of 17 bp[43] (Figure 5A). We showed 6.91% of all possible 6-mers combined with conserved -35 motif can 100% form real promoters (Figure 5B and Supplementary File 4). Meanwhile, 66.2% of all possible 6-mers have more than 50% chances to form real promoters, and these 6-mers contributed to 71.66% chances to form strong promoters (Figure 5B). In comparison,

the 6-mers with less than 50% chances to form real promoters still have 68.8% chances to form strong promoters. Previous study highlighted the contribution of -35 and -10 motifs to promoter function by investigating their binding against RNA polymerase (RNAP)/sigma70[26]. We showed 70% of the -10 motifs with the strongest binding against RNAP/sigma70 have 95% chance to form real promoters, whereas 55% of the weakest binding ones with less than 50% chance to form real promoters (Figure 5C). The strongest binding -10 motifs showed slightly higher chances to form strong promoters (71.65%) than the weakest binding ones (66.75%).

Next, the possible consist of the -35 motif was tested by maintaining the conserved -10 motif (TATAAT). This time, all generated random promoters (including random 6-mers) with TATAAT at the -35 position were predicted as real promoters, and only 8.54% of TATAAT in combination with other possible 6-mers were predicted as weak promoters (Supplementary File 4). However, PromoR predicted real promoter portion decreased obviously while changing TATAAT to other 6-mers.

Please see Line 259-283 in Page 11 in the revised manuscript.

6:

Figure 4D, this figure can reproduced according to promoter activity as Figure 5A and Figure 5E

Response: We are sorry for the confusion.

The figure has been reproduced as below:

7:

The word "de novo" should be either with or without italics, please check the whole manuscript.

Response: We are sorry for the confusion.

We have fixed this error throughout the manuscript.

8:

Line 225, "but promoter strength prediction was hard to achieve especially we adopted high copy number plasmids". This is confused.

Response: We are sorry for the confusion.

The sentence was removed to avoid confusions.

9:

Line 245, "Based on PromoR prediction, 23.5% of all possible 6-mers in the -10 position assembled with -35 motif (TTGAGC) have more than 90% chance to form active promoters, while 100% of all possible 6-mers in the -35 position assembled with -10 motif (TATAAT) can form active promoters (Supplementary File 1)". This sentence is confused.

Response: We are sorry for the confusion.

This sentence was rewritten as below:

Based on PromoR prediction, 23.5% of the random sequences carried all possible 6-mers as the -10 motif and the conserved -35 motif (TTGACA) has a 90% chance to form real promoters (Supplementary File 4). In addition, 100% of the random sequences carried all possible 6-mers as the -35 position and the conserved -10 motif (TATAAT) can form real promoters (Supplementary File 4).

10:

Line 341, the formula is clashed.

Response: We are sorry for the confusion.

The formula in the manuscript was reformatted.

11:

Figure 2C, this table should be better formatted, the values and the headings should be centered.

Response: We are sorry for the confusion.

The Table has been revised as below:

12:

The figure legend of figure 5, egfp is underlined.

Response: We are sorry for the confusion.

The mistake has been revised.

Reviewer #5:

In the reviewed manuscript, authors developed supervised deep learning(DL) models by optimizing previous models for promoter recognition and strength prediction. These models were used for evaluating the key elements for architecting active promoters and investigating factors that affect promoter strength. For promoter design, a diffusion model was adopted for generating de novo promoters followed by predicting promoter activity.

The novel and original results of this work demonstrated that:

1. This study showed DL tools can be implemented for characterizing promoter functions and guiding promoter design.
2. DL tools are more robust for promoter recognition than promoter strength prediction.
3. The DL tools architected can assist the design of novel promoters and provide guidance for the other related networks.

Response: Many thanks to your kindly positive comments.

1.

The title of the manuscript does reflect its content. Abstract is brief enough, indicates the purpose of the work, and includes brief presentation of experiments and main results. Introduction gives some important information concerned to the problem of models for promoter recognition and strength prediction based on relevant literature and clearly

presents rationale and the goal of the study.

Response: Thanks for your suggestion. The title of this article has been changed to “Deep learning-assisted designing novel promoters in *Escherichia coli*” according to your suggestion.

The methods applied in the research part of the manuscript are appropriate and presents enough details allowing repeating of the experiments. Coverage of the subject is complete and well organized. There are relationships to the earlier research given in details. The figures included in the manuscript are justified.

In general, the manuscript presents original and valuable approach to the models for promoter recognition and strength prediction. The authors present good practical experience in promoters design, recognition and strength prediction and provide guidance for the other related networks.

In conclusion, I would recommend the paper for publication in the Advanced Genetics.

Response: Thanks again for your positive comments.

2 nd Peer Review	29-Jul-2023 to 14-Aug-2023
----------------------------

REVIEWER REPORT:

Reviewer #1: This manuscript is about generating novel sigma70 promoters in *Escherichia coli*. The authors developed a computational pipeline, aiming to generate novel *E. coli* promoters with high activation activities. The workflow of this new pipeline consists of three sequential parts: PromoDiff, DRSA design, and Ndesign. To generate novel sigma70 promoters, PromoDiff was developed by taking advantage of the diffusion model, which is integrated by using the UNet architecture. PromoDiff was designed to generate constraint-based sigma70 promoters, maintaining the -35 and -10 motifs. To assess the real/fake, strength, and expression levels of the constraint-based novel sigma70 promoters, PromoR, PromoS, and PromoA were built, respectively. To further improve the prediction of promoter activity, the authors built a simple CNN model, PromoNet/Ndesign.

To assess the performance the pipeline, the authors experimentally validated 200 synthesized promoters, generated only by the first two parts (PromoDiff and DRSA design), and the authors noted that 99.1% of them were active. Additionally, the authors used the 200 promoters as the training data to develop Ndesign, attempting to better predict the promoter activity. To assess the performance of Ndesign, a dataset of 50 constraint-based random promoters was collected and used to predict their activities, and the PCC was 0.4306 when compared with the experimental result.

After a certain extent of revision made by the authors, the design rationale and the novelty of this study are still confusing. I have major concerns on various issues listed below that need to be addressed and then revised accordingly before it can be considered suitable for publication.

Q1. Novelty of this generative pipeline addressed in this manuscript must be clarified.

Generating novel sigma70 promoters, as well as promoter (real/fake) classification, promoter activity and level prediction, are not new topics. As far as I could tell from the current layout of the manuscript, the generative task of constraint-based sigma70 promoters might be a new topic. However, it is vital that the author must indicate how PromoDiff, in terms of the functionality of synthesizing novel promoters, differs from the available methods, such as GAN-based models. The GAN based models could not be

engineered to perform constraint-based promoter generation? Very difficult? Or it is just that GAN based approaches have not been assessed conditioned on the -35/-10 motif constraints?

Q2. Novelty of DRSA design needs to be clarified.

The authors built PromoR/S/A, by using the ResNet architecture along with self-attention layers, to assess the real/fake, strength, and expression levels of the constraint-based novel sigma70 promoters. However, the authors do not mention which ResNet is used, 18, 50, or 101? Is it necessary to use such a deep network architecture? It is quite possible that a simple CNN equipped with only a few convolutional layers may suffice, as also implied by the "...a simple network named PromoNet..." (Page 11, Line 248). For example, in doi:10.1371/journal.pone.0171410, a model consisting of only a convolutional layer can achieve 90% sensitivity and >90% specificity in the classification of E. coli sigma70 promoters. Likewise, it is also possible that iPromoter-CLA can take the place of PromoS, and it can be further tuned to perform the task of PromoA.

Briefly, the authors do not explicitly show the evidence to support that these new models built in this study outperform the available models such as CNNProm, iPromoter-CLA, etc. A table that can allow the audience to easily compare the accuracy, sensitivity, specificity, F1 score, PCC, etc. of PromoR/S/A and of the other available models must be prepared.

If the performance of PromoR/S/A is not obviously superior to that of the other available models, then the novelty of this study should not be attributed to DRSA design.

Q3. Please clarify the contribution of this study, compared to other generative models of sigma70 promoters.

It is still confusing whether this new generative and filtering pipeline really outperforms other generative models. Just aligning the promoter activity to that of J23119 does not appear to provide sufficient evidence.

Q4. Title of this manuscript - revised to reveal the limitations to sigma70 promoters?

It turns out that this study aims at generating novel sigma70 promoters, but not other non-sigma promoters.

Q5. Why taking 50 bp as the default length for the model training? Why not 81 bp as other papers did?

To respond to Q2 raised by Reviewer 3, the authors stated that "Training the model of PromoS with a sequence length of 50 bp is for fitting the generated promoters". Please explain why the generative model cannot be trained to give promoters of 81bps. For example, ref. 29 (iPromoter-CLA) also uses 81 bp promoters retrieved from RegulonDB as the training data.

Q6. The hyperparameters of the models built in this study must be provided

For example, how many convolutional layers are used in PromoNet/Ndesign? What are the kernel and stride sizes used in different layers? Likewise, these hyperparameters of the other models being trained in this study must be clearly listed.

Q7. Quite a few statements are confusing.

Page 4, Line 90: ... however, the combinatory uses to maximize the capability of DL methods...

"the combinatory uses" means GAN-based model, or the new models as created in this study? What does "combinatory" refer to? Why this "uses" needs further researches?

Page 4, Line 72, 73: Due to the high complexity of ...

How we know or define there is "high complexity"?

The authors must know that the two examples given above are like "a drop in the bucket".

Q8. Many verbs used in this manuscript do not conform to the widely used ones

Page 5, Line 93: aims to architect tools how "architect" is different from "create" or "develop" or "build"?

Besides, "brought out" have been used for many times. Why not just use "introduce", "design", "create", "develop", "train", or "build"?

Q9. The writing, wording, and phrasing must be rigorously improved. Grammatical errors are widely found in the manuscript.

The text requires extensive and intensive refinement to enhance its clarity and style. The streamline of main text in this manuscript must be intensively adjusted, to clearly state, for each part of this pipeline, the specific aim, the designing rationale, and the pros and cons based on the benchmarks.

Reviewer #2: The authors have sufficiently addressed my original concerns. I still have concerns that the machine learning is not particularly accurate but I still see it as an important step in an important area of research, so hopefully the work of these authors will spur on further developments to improve these models.

Reviewer #3: The author have answered all my comments.

Second Editorial Decision 14-Aug-2023	
Editorial Decision: Major revision	
Recommendation of the reviewers	
Reviewer #1 Recommends Major Revision	
Reviewer #2 Publish without Revision	
Reviewer #3 Publish without Revision	

Authors' response to 2 nd Review	24-Aug-2023
---	-------------

Update the manuscript based on reviewer #1's comments.

3 rd Peer Review	24-Aug-2023 to 07-Sep-2023
----------------------------

Reviewer #1: This manuscript is about generating novel sigma70 promoters in Escherichia coli. The authors developed a computational pipeline, aiming to generate novel E. coli sigma70 promoters with high activation activities. The workflow of this new pipeline consists of three sequential parts:

PromoDiff, DRSAdesign, and Ndesign. To generate novel sigma70 promoters, PromoDiff was developed by taking advantage of the diffusion model, which has been integrated by using the UNet architecture. PromoDiff was designed to generate constraint-based sigma70 promoters, maintaining the -35 and -10 motifs. To assess the real/fake, strength, and expression levels of the constraint-based novel sigma70 promoters, PromoR, PromoS, and PromoA were built, respectively. To further improve the prediction of promoter activity, the authors built a simple CNN model, PromoNet/Ndesign.

To assess the performance the pipeline, the authors experimentally validated 200 synthesized promoters, generated only by the first two parts (PromoDiff and DRSAdesign), and the authors noted that 99.1% of them were active. Additionally, the authors used the 200 promoters as the training data to develop Ndesign, attempting to better predict the promoter activity. To assess the performance of Ndesign, a dataset of 50 constraint-based random promoters was collected and used to predict their activities, and the PCC was 0.4306 when compared with the experimental result. A notable contribution of this study is the authors' successful use of this pipeline to generate a number of novel sigma70 promoters with strong activity.

Given that I am unable to find the point-to-point response that should have been prepared by the authors in the second revision, the review process for the second revision required much more effort than initially expected. After reviewing this revision, I acknowledge in the main text the enhancements made to the study's design rationale and novelty. However, several concerns persist, as outlined briefly below, that need to be addressed. These concerns must be resolved in the subsequent revision for the manuscript to be deemed fit for publication.

- The quality of the English writing needs significant improvement.

Below are the concerns related to the English writing identified in the abstract:

- * "content of controllable designing" might be better phrased as "the potential for controllably designing".
- * The first mention of an organism should provide its full binomial nomenclature. It should be "Escherichia coli (E. coli) promoter".
- * "brought out" is not a standard phrasing in scientific literature. "Introduced" or "developed" would be more appropriate.
- * "but most promoters with relative low activity" should be "but most promoters had relatively low activity".
- * "architected" is not a common term in this context. "Designed" would be more appropriate.
- * "was brought out relied on" is not appropriate. Consider "Next, we introduced 'Ndesign', which relied on generating...". Besides, "carried" is not the right word choice here. "that carried" or "incorporating" would be better.
- * "advantages" should be "advantage", making it "Taking advantage of".

Below are the concerns related to the English writing identified in the third paragraph of "Introduction":

- * "controllable designing": It might be better to use "controllable design".
- * "methods relied on" should be "methods that rely on" or "methods relying on".

- * "Growing number of DL methods were built" should be "A growing number of DL methods have been developed" for clarity and grammatical accuracy.
- * "generating of novel promoters" "generating of" is not standard English. "generation of" would be correct.
- * "many of the architected models" "architected" is not a standard term in this context. "Designed" or "developed" would be more appropriate.
- * "using an generative adversarial network (GAN)": "an" should be "a" since "generative" starts with a 'g'.
- * "natural existed" should be "naturally existing".
- * "artificial designing" could be changed to "artificially designing" or "the artificial design of".
- * "better generating of" could be changed to "better generation of". Also, "promoter" should be "promoters" to maintain plurality.

The authors must note that the errors mentioned above are only a subset of the potential issues. The revised manuscript has many other issues pertaining to English writing.

- If the primary objective of this manuscript has transitioned to the generation of novel σ_{70} promoters, wouldn't it be more appropriate to train and validate PromoR/S/A specifically on σ_{70} promoters?

The authors evaluated PromoR in comparison to tools tailored for promoter and non-promoter classification. However, in their pursuit to optimize the generation of novel and robust promoters, they narrowed their approach to σ_{70} promoters using a (σ_{70} -35 and -10 motifs) constraint-based generation strategy. This raises questions about whether refining PromoR/S/A's training specifically on σ_{70} promoters could further enhance the pipeline's performance. Such a viewpoint merits at least some consideration in the discussion.

- Please note that the citation format should be consistent throughout the manuscript.

For example, see Line 176, ".[20]", Line 188, ".[29]", and Line 173, "[33]."

- In Figure S1. "16 different nucleotides" should be "16 different di-nucleotides"

Furthermore, the authors did not address several concerns I highlighted in my prior review, neither through direct responses nor modifications in the main text. Consequently, I reiterate them below.

- Title of this manuscript - revised to reveal the limitations to σ_{70} promoters?

It turns out that this study eventually achieved the generation of novel σ_{70} promoters, but not other non- σ_{70} promoters.

- Why taking 50 bp as the default length for the model training? Why not 81 bp as other papers did?

The authors stated that "Training the model of PromoS with a sequence length of 50 bp is for fitting the

generated promoters". Please explain why 50-bp promoters but not 80-bp nor 81-bp promoters have been generated. For example, ref. 29 (iPromoter-CLA) also uses 81 bp promoters retrieved from RegulonDB as the training and validation data.

Third Editorial Decision 07-Sep-2023	
Editorial Decision: Minor revision	
Recommendation of the reviewers	
Reviewer #1 Recommends Minor Revision	

Authors' response to 3 rd Review	21-Sep-2023
---	-------------

Reviewer #1:

Reviewer #1: This manuscript is about generating novel sigma70 promoters in Escherichia coli. The authors developed a computational pipeline, aiming to generate novel E. coli sigma70 promoters with high activation activities. The workflow of this new pipeline consists of three sequential parts: PromoDiff, DRSAdesign, and Ndesign. To generate novel sigma70 promoters, PromoDiff was developed by taking advantage of the diffusion model, which has been integrated by using the UNet architecture. PromoDiff was designed to generate constraint-based sigma70 promoters, maintaining the -35 and -10 motifs. To assess the real/fake, strength, and expression levels of the constraintbased novel sigma70 promoters, PromoR, PromoS, and PromoA were built, respectively. To further improve the prediction of promoter activity, the authors built a simple CNN model, PromoNet/Ndesign.

To assess the performance the pipeline, the authors experimentally validated 200 synthesized promoters, generated only by the first two parts (PromoDiff and DRSAdesign), and the authors noted that 99.1% of them were active. Additionally, the authors used the 200 promoters as the training data to develop Ndesign, attempting to better predict the promoter activity. To assess the performance of Ndesign, a dataset of 50 constraint-based random promoters was collected and used to predict their activities, and the PCC was 0.4306 when compared with the experimental result. A notable contribution of this study is the authors' successful use of this pipeline to generate a number of novel sigma70 promoters with strong activity.

Given that I am unable to find the point-to-point response that should have been prepared by the authors in the second revision, the review process for the second revision required much more effort than initially expected. After reviewing this revision, I acknowledge in the main text the enhancements made to the study's design rationale and novelty. However, several concerns persist, as outlined briefly below, that need to be addressed. These concerns must be resolved in the subsequent revision for the manuscript to be deemed fit for publication.

Response: Thank you for your suggestions. We have carefully read all the comments and provided a point-by-point response. The whole manuscript has been revised as suggested and highlighted by referring to the manuscript preparation checklist.

Response Letter to Reviewers

- The quality of the English writing needs significant improvement.

Response: Thank you for your suggestions. To address the English writing limitations, we have sent the manuscript to a native English speaker and a researcher for providing help.

Below are the concerns related to the English writing identified in the abstract:

* "content of controllable designing" might be better phrased as "the potential for controllably

designing".

Response: Thank you for your suggestions. The phrase has been revised as suggested.

* The first mention of an organism should provide its full binomial nomenclature. It should be "Escherichia coli (E. coli) promoter".

Response: Thank you for your suggestions. The phrase has been revised as suggested.

* "brought out" is not a standard phrasing in scientific literature. "Introduced" or "developed" would be more appropriate.

Response: Thank you for your suggestions. The phrase "brought out" were fully replaced by "Introduced" or "developed" in our manuscript.

* "but most promoters with relative low activity" should be "but most promoters had relatively low activity".

Response: Thank you for your suggestions. The phrase has been revised as suggested.

* "architected" is not a common term in this context. "Designed" would be more appropriate.

Response: Thank you for your suggestions. The phrase has been revised as suggested.

* "was brought out relied on" is not appropriate. Consider "Next, we introduced 'Ndesign', which relied on generating...". Besides, "carried" is not the right word choice here. "that carried" or "incorporating" would be better.

Response: Thank you for your suggestions. The phrase has been revised as suggested.

* "advantages" should be "advantage", making it "Taking advantage of".

Response: Thank you for your suggestions. The phrase has been revised as suggested.

Below are the concerns related to the English writing identified in the third paragraph of "Introduction":

* "controllable designing": It might be better to use "controllable design".

Response: Thank you for your suggestions. The phrase has been revised as suggested.

* "methods relied on" should be "methods that rely on" or "methods relying on".

Response: Thank you for your suggestions. The phrase has been revised as suggested.

* "Growing number of DL methods were built" should be "A growing number of DL methods have been developed" for clarity and grammatical accuracy.

Response: Thank you for your suggestions. The phrase has been revised as suggested.

* "generating of novel promoters" "generating of" is not standard English. "generation of" would be correct.

Response: Thank you for your suggestions. The phrase has been revised as suggested.

* "many of the architected models" "architected" is not a standard term in this context. "Designed" or "developed" would be more appropriate.

Response: Thank you for your suggestions. The phrase "architected" were fully replaced by "Designed" or "developed" in our manuscript.

* "using an generative adversarial network (GAN)": "an" should be "a" since "generative" starts with a 'g'.

Response: Thank you for your suggestions. The phrase "architected" were fully replaced by "Designed" or "developed" in our manuscript.

* "natural existed" should be "naturally existing".

Response: Thank you for your suggestions. The phrase "natural existed" were fully replaced by "naturally existing" in our manuscript.

* "artificial designing" could be changed to "artificially designing" or "the artificial design of".

Response: Thank you for your suggestions. The phrase has been revised as suggested.

* "better generating of" could be changed to "better generation of". Also, "promoter" should be "promoters" to maintain plurality.

Response: Thank you for your suggestions. The phrase has been revised as suggested.

The authors must note that the errors mentioned above are only a subset of the potential issues. The revised manuscript has many other issues pertaining to English writing.

Response: Thank you for your suggestions. The English writing errors were fixed thoroughly.

- If the primary objective of this manuscript has transitioned to the generation of novel sigma70 promoters, wouldn't it be more appropriate to train and validate promoR/S/A specifically on sigma70 promoters?

The authors evaluated PromoR in comparison to tools tailored for promoter and non-promoter classification. However, in their pursuit to optimize the generation of novel and robust promoters, they narrowed their approach to sigma70 promoters using a (sigma70 -35 and -10 motifs) constraintbased generation strategy. This raises questions about whether refining PromoR/S/A's training

specifically on sigma70 promoters could further enhance the pipeline's performance. Such a viewpoint merits at least some consideration in the discussion.

Response: Thank you for your suggestions.

We have added changes to discussion part, as follows,

The models trained with NDB showed less correlation between predicted and real promoter activity[5], which inspired us to organize data with distinctive features for model training. Sigma70 promoters have characteristic sequence elements and account for the major portion in the training sets used in this study. Moreover, PromoS and PromoA displayed high accuracy against sigma70 promoters. Therefore, we considered to generate random promoters carrying motifs of sigma70 promoters to train the PromoS- and PromoA-derived network PromoNet. For the regression prediction, PromoNet achieved a PCC of 0.4946, which is higher than that of the model trained using NDB (0.31).

- Please note that the citation format should be consistent throughout the manuscript.

For example, see Line 176, ".[20]", Line 188, ".[29]", and Line 173, "[33]."

Response: Thank you for your suggestions. The citation format has been revised as suggested.

- In Figure S1. "16 different nucleotides" should be "16 different di-nucleotides"

Response: Thank you for your suggestions. The phrase has been revised as suggested.

Furthermore, the authors did not address several concerns I highlighted in my prior review, neither through direct responses nor modifications in the main text. Consequently, I reiterate them below.

- Title of this manuscript - revised to reveal the limitations to sigma70 promoters?

It turns out that this study eventually achieved the generation of novel sigma70 promoters, but not other non-sigma70 promoters.

Response: Thank you for your suggestions.

It might have some confusion since we developed two methods for designing novel promoters. We have made a lot of efforts to indicate that we developed DRSAdesign, a diffusion model-based network to learn features from all types of promoters and generate different types of promoters. The diffusion model based method together with the supervised model actually able to generate all types of promoters, this has been

written in our Abstract as below:

Firstly, we created “DRSAdesign”, which relied on diffusion model to generate different types of novel promoters, followed by predicting their real/fake and strength. Generating sigma70 promoters only took a small portion in this study. The reason for developing such a method for generating sigma70 promoter is that sigma70 promoters with more distinctive features, which is easier to be learnt by deep learning model. This would help for designing of promoters with desired activity. In total, the two methods developed in this study can be used to design all types of promoters, and simply sigma70 promoters.

- Why taking 50 bp as the default length for the model training? Why not 81 bp as other papers did? The authors stated that "Training the model of PromoS with a sequence length of 50 bp is for fitting the generated promoters". Please explain why 50-bp promoters but not 80-bp nor 81-bp promoters have been generated. For example, ref. 29 (iPromoter-CLA) also uses 81 bp promoters retrieved from RegulonDB as the training and validation data.

Response: Thank you for your suggestions.

The very initial reason for learning and generating promoters at length of 50 bp is that the promoter sequences deposited in NDB contained promoters at a length of 50 bp. In addition, promoters at this length can be used to learn and generate functional promoters, this has been proved by previous study (doi: 10.1093/nar/gkaa325). It is not the first time to use a promoter at a length of 50 bp for training models (doi: 10.1093/nar/gkaa325).

Moreover, even the promoters in NDB can be extended its length to 81 bp, it is uncertain of whether extended from the upstream or the downstream. Our study proved that retrieving promoters to a length of 50 bp from RegulonDB can be used to train a network with comparable high accuracy. These evidences showed promoters at a length of 50 bp is suitable for training, generating, and validating.

4th Peer Review

21-Sep-2023 to 08-Oct-2023

After reviewing this revision, I acknowledge in the abstract and main text the enhancements made to the English writing, as well as to the study's design rationale and novelty. However, I believe that the quality of the English writing still needs improvement. Above all, the authors must know that the items indicated in the following are only a small subset (at least less than 50%) of all potential issues in the current manuscript layout. The revised manuscript has many other issues pertaining to English writing.

Line 83: A growing number of DL methods have been developed to promote strength prediction and generation of novel promoters.

1. The phrase "to promote strength prediction" is ambiguous. It's unclear if "promote" is being used as a verb related to enhancing the prediction or if it relates to "promoter strength prediction".

2. Please change to this: "A growing number of Deep Learning (DL) methods have been developed for predicting promoter strength and generating novel promoters."

Line 248: Our results indicated that 99.1% of the randomly generated promoters were active; two promoters displayed only 17.1% and 11.5% higher FI/OD600 values than the negative control and were hence considered as non-active promoters (Figure 4A).

1. The sentence starts by saying that 99.1% of the randomly generated promoters were active. However,

it then describes two promoters with specific higher FI/OD600 values than the negative control. The relationship between these two statements isn't clear.

2. If the two promoters displayed higher values than the negative control, it's counterintuitive to then label them as "non-active". Typically, higher values than a negative control would imply activity, not the absence of it.

3. The semicolon in the sentence can be replaced with a comma for better flow.

Line 254: Of the promoters with higher activity than J23119, 52% had a spacer length of 16 bp, including the strongest promoter P1, while the remaining 48% had a spacer length of 17 bp. Moreover, 60% of the promoters that exhibited activity values of more than 50% of that of J23119 also had a spacer length of 16 bp, suggesting 16 bp can be an optimal spacer length.

1. The sentence mentions that 52% of the promoters had a spacer length of 16 bp and 48% had a spacer length of 17 bp, which adds up to 100%. This implies that there are no other spacer lengths with higher activity than J23119. So, 18-bp or 15-bp spacer was not permitted/included in the whole assessment of constraint-based design/generation?

2. The phrase "suggesting 16 bp can be an optimal spacer length" might be seen as jumping to a conclusion based solely on the presented data. Without more comprehensive data or a broader context, this assertion may be too strong.

Line 283: Random promoters were generated using an optimal spacer length of 17 bp [43].

As raised by the sentences around Line 254, 16 bp can be optimal. Why 17 bp was still used here? I guess that the authors think the optimal spacer could be either 16bp or 17bp, which should be explicitly indicated in the main text.

=====
- Many verbs in sentences have tense misuse issues.
=====

The authors are advised that in the following I just indicate one of the many potential cases.

Line 263: To generate a model that can accurately predict promoter activity, a simple network named PromoNet was built by only integrating CNN (Figure 4C). PromoNet learns features from the 200 constraint-based randomly generated promoters, which displayed a PCC of 0.4946 using five-fold cross-validation.

1. In the second sentence, "PromoNet learns features from the 200 constraint-based randomly generated promoters," the present tense verb "learns" might be inconsistent with the past tense context provided by the previous sentence.

2. The phrase "which displayed a PCC of 0.4946 using five-fold cross-validation" uses the past tense verb "displayed." Given the rest of the context, it might be clearer to use the present tense or another structure.

=====
- The styles in the References section are not consistently used
=====

See Ref. 3, 4, 9, 10, ... etc. no page or article numbers indicated in respective journal volumes. It becomes challenging for readers to locate the exact sources or content being referred to. It's essential to provide complete citation details to ensure clarity and enable readers to verify the referenced information.

Editorial Decision: Minor revision Recommendation of the reviewers Reviewer #1 Recommends Minor Revision
--

Authors' response to 4 th Review

09-Oct-2023

Reviewer #1:

After reviewing this revision, I acknowledge in the abstract and main text the enhancements made to the English writing, as well as to the study's design rationale and novelty. However, I believe that the quality of the English writing still needs improvement. Above all, the authors must know that the items indicated in the following are only a small subset (at least less than 50%) of all potential issues in the current manuscript layout. The revised manuscript has many other issues pertaining to English writing.

Response: Thank you for your suggestions. We have carefully read all the comments and provided a point-by-point response. The whole manuscript has been revised as suggested and highlighted by referring to the manuscript preparation checklist.

Line 83: A growing number of DL methods have been developed to promote strength prediction and generation of novel promoters.

1. The phrase "to promote strength prediction" is ambiguous. It's unclear if "promote" is being used as a verb related to enhancing the prediction or if it relates to "promoter strength prediction".
2. Please change to this: "A growing number of Deep Learning (DL) methods have been developed for predicting promoter strength and generating novel promoters."

Response: Thank you for your suggestions. The sentence has been revised as suggested.

Line 248: Our results indicated that 99.1% of the randomly generated promoters were active; two promoters displayed only 17.1% and 11.5% higher FI/OD600 values than the negative control and were hence considered as non-active promoters (Figure 4A).

1. The sentence starts by saying that 99.1% of the randomly generated promoters were active. However, it then describes two promoters with specific higher FI/OD600 values than the negative control. The relationship between these two statements isn't clear.

Response Letter to Reviewers

2. If the two promoters displayed higher values than the negative control, it's counterintuitive to then label them as "non-active". Typically, higher values than a negative control would imply activity, not the absence of it.
3. The semicolon in the sentence can be replaced with a comma for better flow.

Response: Sorry for the confusion. The sentence has been revised as below:

Our results indicated that 99.1% of the randomly generated promoters were active, with more than 20% higher FI/OD600 values than the negative control. By contrast, two promoters displayed only 17.1% and 11.5% higher FI/OD600 values than the negative control and were hence considered as inactive promoters (Figure 4A).

Line 254: Of the promoters with higher activity than J23119, 52% had a spacer length of 16 bp, including the strongest promoter P1, while the remaining 48% had a spacer length of 17 bp.

Moreover, 60% of the promoters that exhibited activity values of more than 50% of that of J23119 also had a spacer length of 16 bp, suggesting 16 bp can be an optimal spacer length.

1. The sentence mentions that 52% of the promoters had a spacer length of 16 bp and 48% had a spacer length of 17 bp, which adds up to 100%. This implies that there are no other spacer lengths

with higher activity than J23119. So, 18-bp or 15-bp spacer was not permitted/included in the whole assessment of constraint-based design/generation?

Response: Sorry for the confusion. We have added a description for spacer length as below:

In the present study, random promoters were generated by maintaining the conserved -35 (TTGACA) and -10 (TATAAT) motifs of the sigma70 promoter and the spacer length varied from 16 to 18 bp. Line 240

2. The phrase "suggesting 16 bp can be an optimal spacer length" might be seen as jumping to a conclusion based solely on the presented data. Without more comprehensive data or a broader context, this assertion may be too strong.

Response: Thank you for your suggestions. The sentence has been revised as below: suggesting 16 bp may be an optimal spacer length according to the results in this study. Line 255

Line 283: Random promoters were generated using an optimal spacer length of 17 bp [43]. As raised by the sentences around Line 254, 16 bp can be optimal. Why 17 bp was still used here? I guess that the authors think the optimal spacer could be either 16bp or 17bp, which should be explicitly indicated in the main text.

Response: Sorry for the confusion. The sentence has been revised as below: Random promoters were generated using an optimal spacer length of 17 bp according to the previous study^[43] (Figure 5A).

- Many verbs in sentences have tense misuse issues.

The authors are advised that in the following I just indicate one of the many potential cases. Line 263: To generate a model that can accurately predict promoter activity, a simple network named PromoNet was built by only integrating CNN (Figure 4C). PromoNet learns features from the 200 constraint-based randomly generated promoters, which displayed a PCC of 0.4946 using five-fold cross-validation.

1. In the second sentence, "PromoNet learns features from the 200 constraint-based randomly generated promoters," the present tense verb "learns" might be inconsistent with the past tense context provided by the previous sentence.

Response: Thank you for your suggestions. The sentence has been revised as suggested.

2. The phrase "which displayed a PCC of 0.4946 using five-fold cross-validation" uses the past tense verb "displayed." Given the rest of the context, it might be clearer to use the present tense or another structure.

Response: Thank you for your suggestions. The present tense was used as suggested.

- The styles in the References section are not consistently used

See Ref. 3, 4, 9, 10, ... etc. no page or article numbers indicated in respective journal volumes. It becomes challenging for readers to locate the exact sources or content being referred to. It's essential to provide complete citation details to ensure clarity and enable readers to verify the referenced information.

Response: Thank you for your suggestions. All the references in this manuscript has been revised as suggested.

Final Decision

13-Oct-2023

Accept the revised version for publication as the authors satisfactorily addressed the final comments of the reviewer.